**The Education and Research 3D Radiative Transfer Toolbox (EaR³T) – Towards the Mitigation of 3D Bias in Airborne and Spaceborne Passive Imagery Cloud Retrievals**

Hong Chen[1,2], K. Sebastian Schmidt[1,2], Steven T. Massie[2], Vikas Nataraja[2], Matthew S. Norgren[2], Jake J. Gristey[3,4], Graham Feingold[4], Robert E. Holz[5], Hironobu Iwabuchi[6]

[1]Department of Atmospheric and Oceanic Sciences, University of Colorado, Boulder, CO, USA

[2]Laboratory for Atmospheric and Space Physics, University of Colorado, Boulder, CO, USA

[3]Cooperative Institute for Research in Environmental Sciences, University of Colorado, Boulder, CO, USA

[4]NOAA Chemical Sciences Laboratory, Boulder, CO, USA

[5]Space Science and Engineering Center, University of Wisconsin–Madison, Madison, WI, USA

[6]Center for Atmospheric and Oceanic Studies, Tohoku University, Sendai, Miyagi, Japan

*Correspondence to*: Hong Chen (hong.chen-1@colorado.edu)

**Abstract**

We introduce the Education and Research 3D Radiative Transfer Toolbox (EaR$^3$T, pronounced [ɜ:t]) for quantifying and mitigating artifacts in atmospheric radiation science algorithms due to spatially inhomogeneous clouds and surfaces, and show the benefits of automated, realistic radiance and irradiance generation along extended satellite orbits, flight tracks from entire aircraft field missions, and synthetic data generation from model data. EaR$^3$T is a modularized Python package that provides high-level interfaces to automate the process of 3D radiative transfer (RT) calculations. After introducing the package, we present initial findings from four applications, which are intended as blueprints to future in-depth scientific studies. The first two applications use EaR$^3$T as a satellite radiance simulator for the NASA Orbiting Carbon Observatory 2 (OCO-2) and Moderate Resolution Imaging Spectroradiometer (MODIS) missions, which generate synthetic satellite observations with 3D-RT on the basis of cloud field properties from imagery-based retrievals and other input data. In the case of inhomogeneous cloud fields, we show that the synthetic radiances are often inconsistent with the original radiance measurements. This lack of radiance consistency points to biases in heritage imagery cloud retrievals due to sub-pixel resolution clouds and 3D-RT effects. They come to light because the simulator's 3D-RT engine replicates processes in nature that conventional 1D-RT retrievals do not capture. We argue that 3D radiance consistency (closure) can serve as a metric for assessing the performance of a cloud retrieval in presence of spatial cloud inhomogeneity even with limited independent validation data. The other two applications show how airborne measured irradiance data can be used to independently validate imagery-derived cloud products via radiative closure in irradiance. This is accomplished by simulating downwelling irradiance from geostationary cloud retrievals of Advanced Himawari Imager (AHI) along all the below-cloud aircraft flight tracks of the Cloud, Aerosol and Monsoon Processes Philippines Experiment (CAMP$^2$Ex, NASA 2019), and comparing the irradiances with the collocated airborne measurements. In contrast to case studies in the past, EaR$^3$T facilitates the use of observations from entire field campaigns for the statistical validation of satellite-derived irradiance. From the CAMP$^2$Ex mission, we find a low bias of 10% in the satellite-derived cloud transmittance, which we are able to attribute to a combination of the coarse resolution of the geostationary imager and 3D-RT biases. Finally, we apply a recently developed context-aware Convolutional Neural Network (CNN) cloud retrieval framework to high-resolution airborne imagery from CAMP$^2$Ex and show that the retrieved cloud optical thickness fields lead to better 3D radiance consistency than the heritage independent pixel algorithm, opening the door to future mitigation of 3D-RT cloud retrieval biases.

## 1. Introduction

Three-dimensional cloud effects in imagery-derived cloud properties have long been considered an unavoidable error source when estimating the radiative effect of clouds and aerosols. Consequently, research efforts involving satellite, aircraft, and surface observations in conjunction with modeled clouds and radiative transfer calculations have focused on systematic bias quantification under different atmospheric conditions. Barker and Liu (1995) studied the so-called independent pixel approximation (IPA) bias in cloud optical thickness (COT) retrievals from shortwave cloud reflectance. The bias arises when approximating the radiative transfer relating to COT and measured reflectance at the pixel or cloud column level through one-dimensional (1D) radiative transfer (RT) calculations, while ignoring its radiative context. However, net horizontal photon transport and other effects such as shading engender column-to-column radiative interactions that can only be captured in a three-dimensional (3D) framework, and can be regarded as a 3D perturbation or bias relative to the 1D-RT (IPA) baseline. 3D biases affect not only cloud remote sensing but they also propagate into the derived irradiance fields and cloud radiative effects (CRE). Since the derivation of regional and global CRE relies heavily on satellite imagery, any systematic 3D bias impacts the accuracy of the Earth's radiative budget. Likewise, imagery-based aerosol remote sensing in the vicinity of clouds can be biased by net horizontal photon transport (Marshak et al., 2008). Additionally, satellite shortwave spectroscopy retrievals of $CO_2$ mixing ratio are affected by nearby clouds (Massie et al., 2017), albeit through a different physical mechanism than in aerosol and cloud remote sensing.

Given the importance of 3D perturbations for atmospheric remote sensing, ongoing research seeks to mitigate the 3D effects. Cloud tomography, for example, inverts multi-angle radiances to infer the 3D cloud extinction distribution (Levis et al., 2020). This is achieved through iterative adjustments to the cloud field until the calculated radiances match the observations. Convolutional neural networks (CNNs, Masuda et al., 2019; Nataraja et al., 2022) account for 3D-RT perturbations in COT retrievals through pattern-based machine learning that operates on collections of imagery pixels, rather than treating them in isolation like IPA. Unlike tomography, CNNs require training based on extensive cloud-type specific synthetic data with the ground truth of cloud optical properties and their associated radiances from 3D-RT calculations. Once the CNNs are trained, they do not require real-time 3D-RT calculations and can therefore be useful in an operational setting. Whatever the future may hold for context-aware multi-pixel or multi-sensor

cloud retrievals, there is a paradigm shift on the horizon that started when the radiation concept
for the Earth Clouds, Aerosol and Radiation Explorer (EarthCARE, Illingworth et al., 2015) was
first proposed (Barker et al., 2012). It foresees a closure loop where broadband radiances, along
with irradiance, are calculated in a 3D-RT framework from multi-sensor input fields (Barker et al.,
2011), and subsequently compared to independent observations by radiometers pointing in three
directions (nadir, forward-, and backward-viewing along the orbit). This built-in radiance closure
can serve as an accuracy metric for any downstream radiation products such as heating rates and
CRE. Any inconsistencies can be used to nudge the input fields towards the truth in subsequent
loop iterations akin to optimal estimation, or propagated into uncertainties of the cloud and
radiation products.

This general approach to radiative closure is also being considered for the National

Aeronautics and Space Administration (NASA) Atmospheric Observation System (AOS,
developed under the A-CCP, Aerosol and Cloud, Convection and Precipitation study), a mission
that is currently in its early implementation stages. Owing to its focus on studying
aerosol-cloud-precipitation-radiation interactions at the process level, it requires radiation
observables at a finer spatial resolution than achieved with missions to date. At target scales close
to 1 km, 3D-RT effects are much more pronounced than at the traditional 20 km scale of NASA
radiation products (O'Hirok and Gautier, 2005; Ham et al., 2014; Song et al., 2016; Gristey et al.,
2020a). Since this leads to biases beyond the desired accuracy of the radiation products, mitigation
of 3D-RT cloud remote sensing biases needs to be actively pursued over the next few years.

Transitioning to an explicit treatment of 3D-RT in operational approaches entails a new

generation of code architectures that can be easily configured for various instrument constellations,
interlink remote sensing parameters with irradiances, heating rates, and other radiative effects, and
can be used for automated processing of large data quantities. A number of 3D solvers are available
for different purposes, for example, the I3RC (International Intercomparison of 3D Radiation
Codes: Cahalan et al., 2005) community Monte Carlo code[1], which now also includes an online
simulator[2] that was described in Várnai et al. (2022) and used in Gatebe et al. (2021); MCARaTS
(Monte Carlo Atmospheric Radiative Transfer Simulator[3]: Iwabuchi, 2006); MYSTIC (Monte

---

[1] https://earth.gsfc.nasa.gov/climate/model/i3rc, last accessed on 26 November, 2022.
[2] http://i3rcsimulator.umbc.edu, last accessed on 26 November, 2022.
[3] https://sites.google.com/site/mcarats/monte-carlo-atmospheric-radiative-transfer-simulator-mcarats, last accessed on 26 November, 2022.

Carlo code for the physically correct tracing of photons in cloudy atmospheres: Mayer, 2009),
which is embedded in libRadtran (library for radiative transfer, Mayer and Kylling, 2005);
McSCIA (Monte Carlo [RT] for SCIAmachy: Spada et al., 2006), which is optimized for satellite
radiance simulations (including limb-viewing) in a spherical atmosphere; McARTIM
(Deutschmann et al., 2011), with several hyperspectral polarimetric applications such as
differential optical absorption spectroscopy; and SHDOM (Spherical Harmonic Discrete Ordinate
Method[4]: Evans, 1998), which, unlike the other methods, is a deterministic solver with polarimetric
capabilities (Doicu et al., 2013; Emde et al., 2015) that is differentiable and can therefore be used
for tomography (Loveridge et al., 2022).
For the future operational application of 3D-RT, it is, however, desirable to run various
different solvers in one common architecture that automates the processing of various formats of
3D atmospheric input fields (including satellite data), allows the user to choose from various
options for atmospheric absorption and scattering, and simulates radiance and irradiance data for
real-world scenes. Here, we introduce one such tool that could serve as the seed for this architecture:
the Education and Research 3D Radiative Transfer Toolbox (EaR$^3$T, pronounced [ɜːt]). It has been
developed over the past few years at the University of Colorado to automate 3D-RT calculations
based on imagery or model cloud fields. It can be operated in two ways– 1) with minimal user
input, where certain RT parameters are bypassed through default settings, for quick radiation
conceptual analysis; 2) with detailed RT parameters setup by user for radiation closure purpose.
EaR$^3$T is maintained and extended by graduate students as part of their education, and applied to
various different research projects including machine learning for atmospheric radiation and
remote sensing (Gristey et al., 2020b; 2022; Nataraja et al., 2022), as well as radiative closure and
satellite simulators. It is implemented as a modularized Python package with various application
codes that combine the functionality in different ways, which, once set up, autonomously process
large amounts of data required by airborne and satellite remote sensing and for machine learning
applications.
The goal of the paper is to introduce EaR$^3$T as a versatile tool for systematically quantifying
and mitigating 3D cloud effects in radiation science as foreseen in future missions. To do so, we
will first showcase EaR$^3$T as an automated radiance simulator for two satellite instruments, the
Orbiting Carbon Observatory-2 (OCO-2, application code 1, App. 1) and the Moderate Resolution

---

[4] https://coloradolinux.com/shdom, last accessed on 26 November, 2022.

Imaging Spectroradiometer (MODIS, application code 2, App. 2) from publicly available satellite
retrieval products. In the spirit of radiance closure, the intended use is the comparison of modeled
radiances with the original measurements to assess the accuracy of the input data, as follows:
operational IPA COT products are made using 1D-RT, and thus the accompanying radiances are
consistent with the original measurements under that 1D-RT assumption only. That is,
self-consistency is assured if 1D-RT is used in both the inversion and radiance simulation.
However, since nature creates 3D-RT radiation fields, we break this traditional symmetry in this
manuscript and introduce the concept of 3D radiance consistency where closure is only achieved
if the original measurements are consistent with the 3D-RT (rather than the 1D-RT) simulations.
The level of inconsistency is then used as a metric for the magnitude of 3D-RT retrieval artifacts
as envisioned by the architects of the EarthCARE radiation concept (Barker et al., 2012).
Subsequently, we discuss applications where EaR$^3$T performs radiative closure in the
traditional sense, i.e., between irradiances derived from satellite products and collocated airborne
or ground-based observations. The aircraft Cloud, Aerosol and Monsoon Processes Philippines
Experiment (CAMP$^2$Ex, Reid et al., 2023), conducted by NASA in the Philippines in 2019, serves
as a testbed of this approach. Here, we use EaR$^3$T's automated processing capabilities to derive
irradiance from geostationary imagery cloud products and then compare these to cumulative
measurements made along all flight legs of the campaign (application code 3, App. 3). In contrast
to previous studies that often rely on a number of cases (e.g., Schmidt et al., 2010; Kindel et al.,
2010), we perform closure systematically for the entire data set, enabling us to identify 3D-RT
biases in a statistically significant manner. Finally, we apply a regionally and cloud type specific
CNN, introduced by Nataraja et al. (2022) that is included with the EaR$^3$T distribution, to
high-resolution camera imagery from CAMP$^2$Ex. This last example demonstrates mitigation of
3D-RT biases in cloud retrievals using the concept of radiance closure to quantify its performance
against the baseline IPA (application code 4, App. 4).
The general concept of EaR$^3$T with an overview of the applications, along with the data
used for both parts of the paper is presented in section 2, followed by a description of the
procedures of EaR$^3$T in section 3. Results for the OCO-2 and MODIS satellite simulators (part 1)
are shown in section 4, followed by the quantification and mitigation of 3D-RT biases with
CAMP$^2$Ex data in section 5 and section 6 (part 2). A summary and conclusion are provided in
section 7. The code, along with the applications presented in this paper, can be downloaded from
the GitHub repository: https://github.com/hong-chen/er3t.

**2. Functionality and Data Flow within EaR³T**

**2.1 Overview**

To introduce EaR³T as a satellite radiance simulator tool and to demonstrate its use for the
quantification and mitigation of 3D cloud remote sensing biases, five applications (Figure 1) are
included in the GitHub software release:

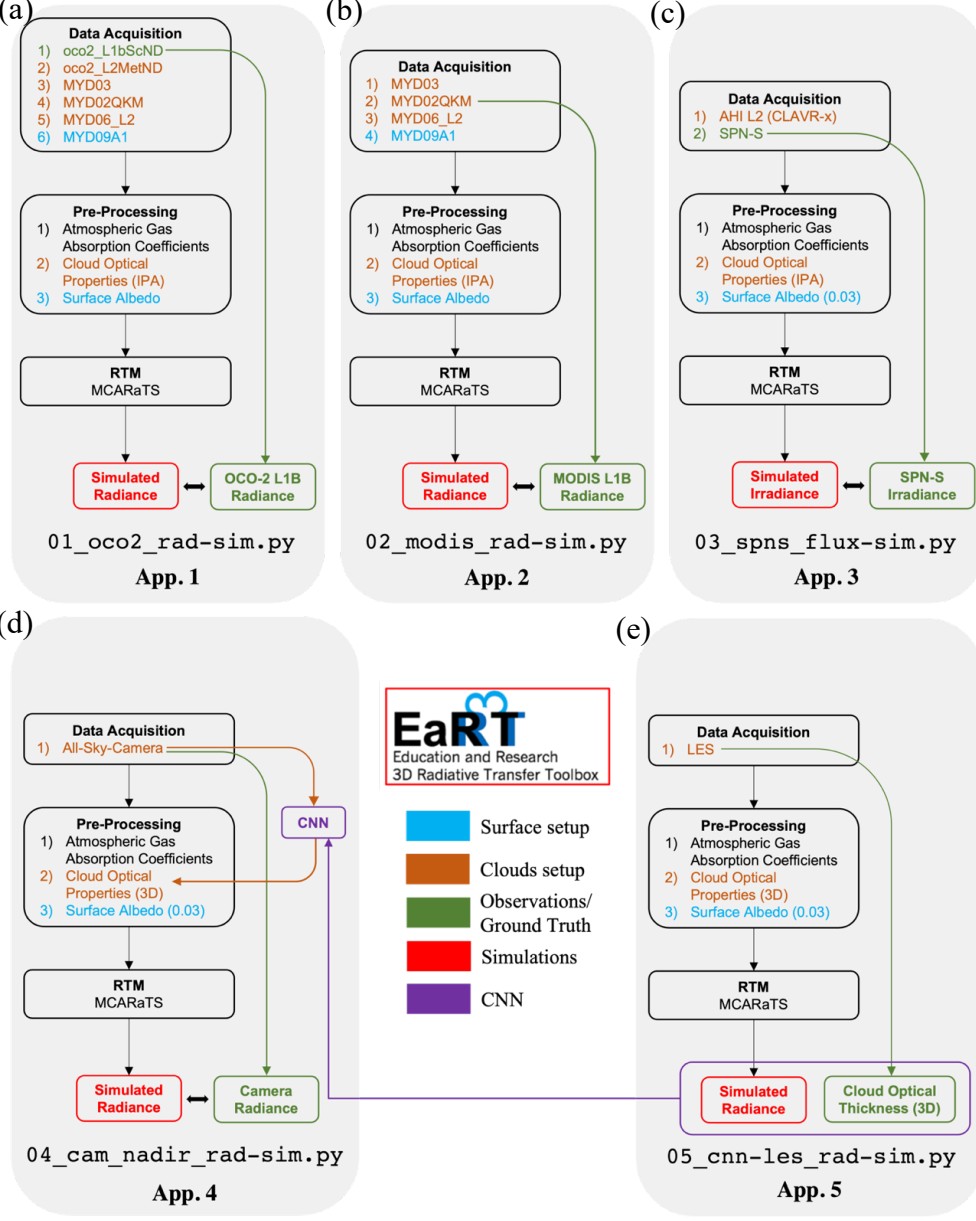


**Figure 1.** Flow charts of EaR³T applications for (a) OCO-2 radiance simulation at 768.52 nm (data described in section 2.2.1 and 2.2.2, results discussed in section 4.1), (b) MODIS radiance simulation at 650 nm (data described in section 2.2.1, results discussed in section 4.2), (c) SPN-S irradiance simulation at 745 nm (data described in section 2.2.3 and 2.2.4, results discussed in section 5), (d) all-sky camera radiance simulation at 600 nm (data described in section 2.2.5, results discussed in section 6), and (e) radiance simulation at 600 nm based on LES data for CNN training (Appendix B). The data products and their abbreviations are described in section 2.2.

1. App. 1, section 4.1 (`examples/01_oco2_rad-sim.py`): Radiance simulations along the track of OCO-2, based on data products from MODIS and others – to assess consistency (closure) between simulated and measured radiance;

2. App. 2, section 4.2 (`examples/02_modis_rad-sim.py`): MODIS radiance simulations – to assess self-consistency of MODIS level-2 (L2) products with the associated radiance fields (L1B product) under spatially inhomogeneous conditions;

3. App. 3, section 5 (`examples/03_spns_flux-sim.py`): Irradiance simulations along aircraft flight tracks, utilizing the L2 cloud products of the AHI, and comparison with aircraft measurements – to quantify retrieval biases due to 3D cloud structure based with data from an entire aircraft field campaign;

4. App. 4, section 6 (`examples/04_cam_nadir_rad-sim.py`): Mitigation of 3D cloud biases in passive imagery COT retrievals from an airborne camera, application of a convolutional neural network (CNN) and subsequent comparison of CNN-derived radiances with the original measurements – to illustrate how the radiance self-consistency concept assesses the fidelity of cloud retrievals.

5. App. 5, Appendix B (`examples/05_cnn-les_rad-sim.py`): Generation of training data for the CNN (App. 4) based on LES inputs. The training datasets contains 1) the ground truth of COT from the LES data; 2) realistic radiance simulated by EaR³T based on the LES cloud fields.

Figure 1 shows the high-level workflow of the applications. The first four share the general concept of evaluating simulations (the output from the EaR³T, indicated in red at the bottom of each column) with observations (indicated in green at the bottom) from various satellite and aircraft instruments. The workflow of each application consists of three parts – 1) data acquisition, 2) pre-processing, and 3) RTM setup and execution. EaR³T includes functions to ingest data from

various different sources, e.g., satellite data from publicly available data archives, which can be
combined in different ways to accommodate input data depending on the application specifics. For
example, in App. 1, EaR$^3$T is used to automatically download and process MODIS and OCO-2
data files based on the user-specified region, date and time. Building on the templates provided in
the current code distribution, the functionality can be extended to new spaceborne or airborne
instruments. Panel (e) of Figure 1 shows a fifth application that was developed for earlier papers
(Gristey et al., 2020a and 2020b; Nataraja et al., 2022; Gristey et al., 2022). In contrast to the first
four, which use imagery products as input, the fifth application ingests model output from a Large
Eddy Simulation (LES) and produces irradiance data for surface energy budget applications, or
synthetic radiance fields for training a CNN. Details and results are described in the respective
papers. The remainder of Section 2 introduces the data used in this paper, as well as the input for
EaR$^3$T. Subsequently, Section 3 describes the EaR$^3$T procedures.

**2.2 Data**

The radiance simulations in App. 1 and App. 2 use data from the OCO-2 and MODIS-Aqua
instruments, both of which are in a sun-synchronous polar orbit with an early-afternoon equator
crossing time within NASA's A-Train satellite constellation. Figure 2 visualizes radiance
measurements by OCO-2 in the context of MODIS Aqua imagery over a partially vegetated and
partially cloud-covered land, illustrating that MODIS provides imagery and scene context for
OCO-2, which in turn observes radiances from a narrow swath. The region is located in southwest
Colorado in the United States of America. We selected this case because both the surface and
clouds are varied along with diverse surface types. The surface features green forest and brown
soil, whereas clouds include small cumulus and large cumulonimbus. In addition, this scene
contains relatively homogeneous cloud fields in the north and inhomogeneous cloud fields in the
south, which allows us to evaluate the simulations from various aspects of cloud morphology. To
simulate the radiances of both instruments we use data products from OCO-2 and MODIS, as well
as reanalysis products from NASA's Global Modeling and Assimilation Office (GMAO) sampled
at OCO-2 footprints and distributed along with OCO-2 data (section 2.2.2).

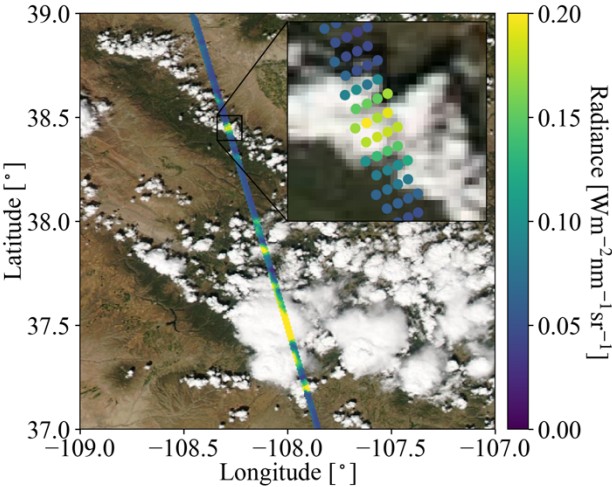

**Figure 2**. OCO-2 measured radiance (units: $Wm^{-2}nm^{-1}sr^{-1}$) at 768.52 nm, overlaid on MODIS Aqua RGB imagery over southwestern Colorado (USA) on 2 September, 2019. The inset shows an enlarged portion along the track, illustrating that OCO-2 radiances co-vary with MODIS-Aqua radiance observations (the circles are used to indicate the geolocation of OCO-2 footprints).

For App. 3 (irradiance simulations and 3D cloud bias quantification), we use geostationary imagery from the Japanese Space Agency's Advanced Himawari Imager to provide cloud information in the area of the flight path of the NASA CAMP$^2$Ex aircraft (Reid et al., 2023). The AHI data are used in conjunction with aircraft measurements of shortwave spectral radiation (section 2.2.4). Subsequently (App. 4: 3D cloud bias mitigation), we demonstrate the concept of radiance closure under partially cloudy conditions with airborne camera imagery (section 2.2.5). The underlying cloud retrieval is based on a convolutional neural network (CNN), which is described in a related paper (Nataraja et al., 2022) in this special issue and relies on EaR$^3$T-generated synthetic radiance data based on Large Eddy Simulations (LES).

### 2.2.1 Moderate Resolution Imaging Spectroradiometer (MODIS)

The MODIS instruments are multi-use multispectral radiometers onboard NASA's Terra and Aqua satellites, which were launched in 1999 and 2002 respectively. MODIS was conceived as a central element of the Earth Observing System (EOS, King and Platnick, 2018). For App. 1 and App. 2, EaR$^3$T ingests MODIS level 1B radiance products at the quarter kilometer scale (channels 1 and 2, bands centered at 650 and 860 nm), MxD02QKM, where 'x' stands for 'O' in the case of MODIS on Terra, and 'Y' in the case of Aqua data), the geolocation product (MxD03), the level 2 cloud product (MxD06), and the surface BRDF (bidirectional reflectance distribution

function) product (MCD43A3). For this paper, we mainly use Aqua data (MYD) from data
collection 6.1.

For cloud properties in App. 2, we use the MODIS cloud product (MxD06L2, collection
6.1). It provides cloud properties such as cloud optical thickness (COT), cloud effective radius
(CER), cloud thermodynamic phase, cloud top height (CTH), etc. (Nakajima and King, 1990;
Platnick et al., 2003). Since 3D cloud effects such as horizontal photon transport are most
significant at small spatial scales (e.g., Song et al., 2016), we use the high-resolution red (650 nm)
channel 1 (250 m), and derive COT directly from the reflectance in the Level-1B data
(MYD02QKM) instead of using the coarser-scale operational product from MYD06. CER and
CTH are sourced from MYD06 and re-gridded to 250 m. The EaR$^3$T strategy for MODIS data is
similar, in principle, to the more advanced method by Deneke et al. (2021), which uses a
high-resolution wide-band visible channel from geostationary imagery to up-sample narrow-band
coarse-resolution channels. However, we simplified cloud detection and COT retrieval (referred
to as COT$_{IPA}$) from reflectance data for the purpose of our paper by using a threshold method
(Appendix C1) and an IPA reflectance-to-COT mapping (Appendix C2). In future versions of
EaR$^3$T this will be upgraded to more sophisticated algorithms. A simple algorithm (Appendix D1)
is used to correct for the parallax shift based on the sensor geometries and cloud heights. The cloud
top height data is provided by the MODIS L2 cloud product and assuming cloud base is the same.

For the surface albedo required by the RTM, we used MCD43A3, which provides BRDF
calculated from a combination of Aqua and Terra MODIS and MISR (Multi-Angle Imaging
Spectroradiometer) clear-sky observations aggregated over a 16-day period (Strahler et al., 1999).
This product contains white sky albedo (WSA, also known as bihemispherical reflectance), which
is obtained by integrating the BRDF over all viewing angles (Strahler et al., 1999). The WSA is
available on a sinusoidal grid with a spatial resolution of 500 m for MODIS band 2, and includes
atmospheric correction for gas and aerosol scattering and absorption. Assuming a Lambertian
surface in this first release of EaR$^3$T, we used the WSA (referred to as surface albedo from now
on) as surface albedo input to the RTM.

**2.2.2 Orbiting Carbon Observatory 2 (OCO-2)**

The OCO-2 satellite was inserted into NASA's A-Train constellation in 2014 and flies
about 6 minutes ahead of Aqua. OCO-2 provides the column-averaged carbon dioxide (CO$_2$)

dry-air mole fraction ($XCO_2$) through passive spectroscopy based on hyperspectral radiance observations in three narrow wavelength regions, the Oxygen A-Band (~0.76 micron), the weak $CO_2$ band (~1.60 micron), and the strong $CO_2$ band (~2.06 micron). As shown in the inset of Figure 2, it takes measurements in eight footprints across a narrow swath. Each of the footprints has a size around 1-2 km, and the spectra for the three bands are provided by separate, co-registered spectrometers (Crisp et al., 2015).

The used OCO-2 data products are 1) Level 1B calibrated and geolocated science radiance spectra (L1bScND), 2) standard Level 2 geolocated $XCO_2$ retrievals results (L2StdND), 3) meteorological parameters interpolated from GMAO (L2MetND) at OCO-2 footprint location. Since MODIS on Aqua overflies a scene 6 minutes after OCO-2, the clouds move with the wind over this time period. We therefore added a wind correction on top of the parallax-corrected cloud fields obtained from MODIS (section 2.2.1). This was done with the 10 m wind speed data from L2MetND (see Appendix D2). For the same scene as shown in Figure 2, Figure 3 shows (a) $COT_{IPA}$, (b) CER, and (c) CTH, all corrected for both parallax and wind effects (these corrections are shown in Figure A5 in Appendix D2). The parallax and wind corrections are imperfect as certain assumptions are involved. For example, they rely on the cloud top height from the MODIS cloud product. In addition, they process the whole scene with one single sensor viewing geometry. To minimize artifacts introduced by the assumptions, one can apply the simulation to a smaller region.

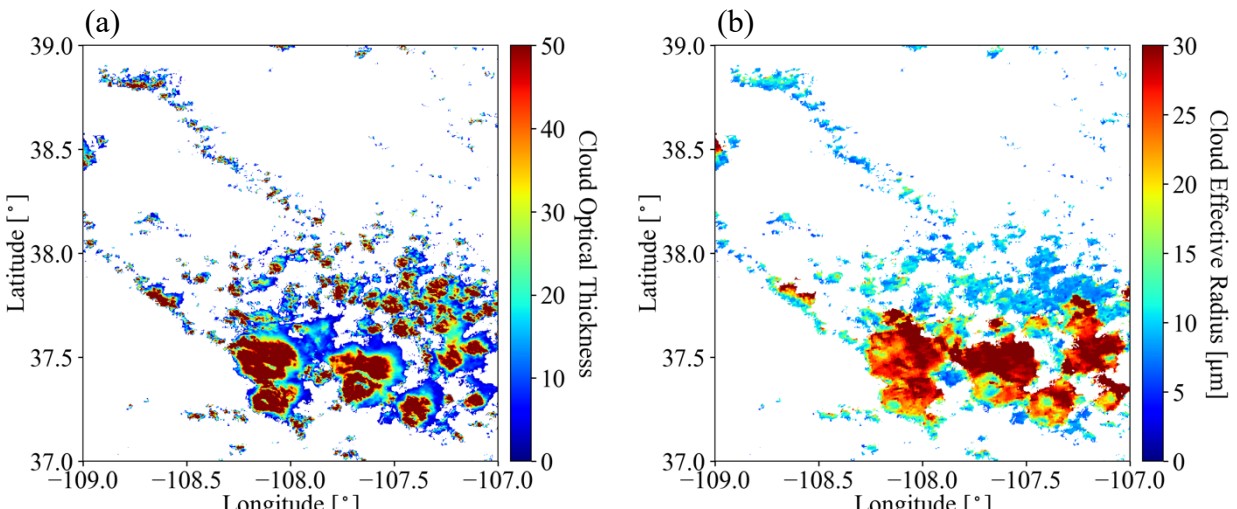

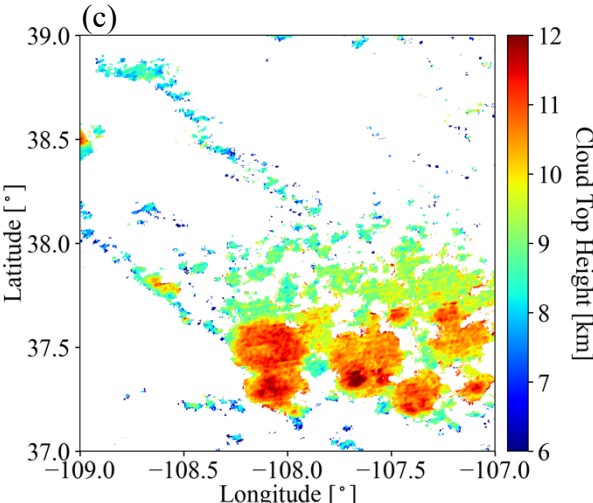


**Figure 3**. **(a)** Cloud optical thickness derived from MODIS L1B radiance at 650 nm by the IPA reflectance-to-COT
mapping (Appendix C2), **(b)** cloud effective radius (units: $\mu$m), and **(c)** cloud top height (units: km)
collocated from the MODIS L2 cloud product. The locations of the cloudy pixels were shifted to account
for parallax and wind effects. The parallax correction ranged from near 0 for low clouds and 1 km for high
clouds (10 km CTH). The wind correction was around 0.8 km, given the median wind speed of 2 m/s to the
east.

325    The OCO-2 data (L2StdND) themselves only provide sparse surface BRDF (referred to as
surface albedo from now on) for the footprints that are clear, while EaR³T requires surface albedo
for the whole domain. Therefore, we used MCD43A3 as a starting point. However, since MODIS
does not have a channel in the Oxygen A-Band, MODIS band 2 (860 nm) was used as a proxy for
the 760 nm OCO-2 channel as follows: we collocated the OCO-2 retrieved 760 nm surface albedo
$\alpha_{OCO}$ within the corresponding 860 nm MODIS MCD43A3 data $\alpha_{MOD}$ as shown in Figure 4a
(same domain as Figures 2 and 3) and calculated a scaling factor assuming a linear relationship
between $\alpha_{OCO}$ and $\alpha_{MOD}$ ( $\alpha_{OCO} = c \cdot \alpha_{MOD}$ ). Figure 4b shows $\alpha_{OCO}$ versus $\alpha_{MOD}$ for all
cloud-free OCO-2 footprints. The red line shows a linear regression (derived scale factor *c=0.867*).
Optionally, the OCO-2-scaled MODIS-derived surface albedo fields can be replaced by the OCO-2
surface albedo products for pixels where they are available. The replacement is done for App. 1.
The scaled and replaced surface albedo is then treated as input to the RTM assuming a Lambertian
surface.

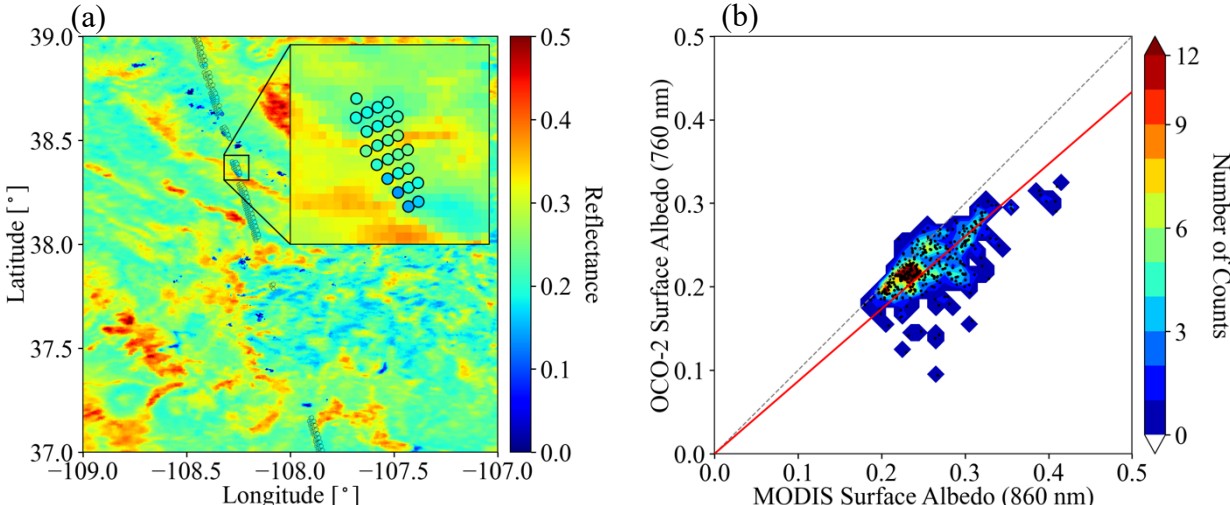

**Figure 4**. **(a)** Surface albedo from the OCO-2 L2 product in the Oxygen A-band (near 760 nm), overlaid on the surface albedo from the MODIS MCD43A3 product at 860 nm. **(b)** OCO-2 surface albedo at 760 nm versus MODIS surface albedo at 860 nm, along with linear regression ($\alpha_{OCO} = c \cdot \alpha_{MOD}$) as indicated by the red line (slope *c=0.867*).

### 2.2.3 Advanced Himawari Imager (AHI)

The Advanced Himawari Imager (AHI, used for App. 3) is a payload on Himawari-8, a geostationary satellite operated by the Meteorological Satellite Center (MSC) of the Japanese Meteorological Agency. The AHI provides 16 channels of spectral radiance measurements from the shortwave (0.47μm) to the infrared (13.3μm). During CAMP$^2$Ex, the NASA in-field operational team closely collaborated with the team from MSC to provide AHI satellite imagery at the highest resolution over the Philippine Sea. From the AHI imagery, the cloud product generation system - Clouds from AVHRR Extended System (CLAVR-x), was used to generate cloud products from the AHI imagery (Heidinger et al., 2014). The cloud products from CLAVR-x include cloud optical thickness, cloud effective radius, and cloud top height at 2 (at nadir) to 5 km spatial resolution. Since AHI provides continuous regional scans every 10 minutes the AHI cloud product has a temporal resolution of 10 minutes.

### 2.2.4 Spectral Sunshine Pyranometer (SPN-S)

The SPN-S is a prototype spectral version of the commercially available global-diffuse SPN1 pyranometer (Wood et al., 2017; Norgren et al., 2022). The radiometer uses a 7-detector design in combination with a fixed shadow mask that enables the simultaneous measurement of both diffuse and global irradiances, from which the direct component of the global irradiance is

calculated via subtraction. The detector measures spectral irradiance from 350 to 1000 nm, and the
spectrum is sampled at 1 nm resolution with 1 Hz timing.
During the CAMP$^2$Ex mission, the SPN-S was mounted to the top of the NASA P-3 aircraft
where it sampled downwelling solar irradiance. To ensure accurate measurements, pre- and
post-mission laboratory-based calibrations were completed using tungsten "FEL" lamps that are
traceable to a National Institute of Standards and Technology standard. Additionally, the direct
and global irradiances were corrected for deviations of the SPN-S sensor plane from horizontal
that are the result of changes in the aircraft's pitch or roll. This attitude correction applied to the
irradiance data is a modified version of the method outlined in Long et al. (2010). However,
whereas Long et al. (2010) employ a "box" flight pattern to characterize the sensor offset angles,
in this study an aggregation of flight data containing aircraft heading changes under clear-sky
conditions are used as a substitute. The estimated uncertainty of the SPN-S system is 6 to 8%, with
4 to 6% uncertainty stemming from the radiometric lamp calibration process, and up to another 2%
resulting from insufficient knowledge of the sensor cosine response. The stability of the system
under operating conditions is 0.5%. A thorough description of the SPN-S and its calibration and
correction procedures is provided in Norgren et al. (2022). In this paper (App. 3) only the global
downwelling irradiance sampled by the 745 nm channel is used.

**2.2.5 Airborne All-Sky Camera (ASC)**
The All-Sky Camera (used for App. 4) is a commercially available camera (ALCOR
ALPHEA 6.0CW[5]) with fish-eye optics for hemispheric imaging. It has a Charge-Coupled Device
(CCD) detector that measures radiances in red, green, and blue channels. Radiometric and
geometric calibrations were performed at the Laboratory of Atmospheric and Space Physics at the
University of Colorado Boulder. The three-color channels are centered at 493, 555, and 626 nm
for blue, green, and red, respectively, with bandwidths of 50 – 100 nm. Only radiance data from
the red channel are used in this paper. The spatial resolution of the ASC depends on the altitude of
the aircraft and the viewing zenith angle. Across the hemispheric field of view of the camera, the
resolution of the field angle is approximately constant, at about 0.09º. At a flight level of 5 km,

---

[5]https://www.alcor-system.com/common/allSky/docs/ALPHEA_Camera%20ALL%20SKY%20CAMERA_Doc.pdf
last accessed on April 24, 2022.

this translates to a spatial resolution of 8 m at nadir. However, due to accuracy limitations of the
geometric calibration and the navigational data from Inertial Navigation System (INS), the nadir
geolocation accuracy could only be verified to within ±50 m. During the CAMP²Ex flights, the
camera exposure time was set manually to minimize saturation of the detector. The standard image
frame rate is 1 Hz. The precision of the camera radiances is on the order of 1%, and the radiometric
accuracy is 6 – 7%.

**3. EaR³T Procedures**
In the previous section, we described the input data for the EaR³T applications. In this
section, we will focus on providing the complete workflow (shown in Figure 1) for the five
applications.
After the required data files have been automatically downloaded in the data acquisition
step as described in previous section, EaR³T pre-processes them and generates the optical
properties of atmospheric gases, clouds, aerosols, and the surface. In Figure 1, the mapping from
input data to these properties is color-coded component-wise (brown for associated cloud property
processing if available, blue for associated surface property processing if available, green for
associated ground truth property). The EaR³T code base used in this paper (v0.1.1; Chen and
Schmidt, 2022) only includes MCARaTS as the 3D RT solver, but others are planned for the future.
MCARaTS is a radiative transfer solver that uses a Monte Carlo photon-tracing method (Iwabuchi,
2006). It outputs radiation (radiance or irradiance) based on the inputs of radiative properties of
surface and atmospheric constituents (e.g., gases, aerosols, clouds) such as single scattering albedo,
scattering phase function or asymmetry parameter, along with solar and sensor viewing geometries.
The setup of these input properties is implemented in EaR³T's pre-processing steps, which
translates atmospheric properties into solver-specific input with minimum user intervention. To
achieve this, EaR³T is modular so that it can be extended as new solvers are added. Although the
five specific applications in this paper do not include aerosol layers, the setup of aerosol fields is
fully supported and has been used in other applications (e.g., Gristey et al., 2022). After pre-
processing, the optical properties are fed into the RT solver. Finally, the user obtains radiation
output from EaR³T, either radiance or irradiance. The output is saved in HDF5 format and can be
easily distributed and accessed by various programming languages. The data variables contained
in the HDF5 output are provided in Table A2 in Appendix A1.

The processes of data acquisition, pre-processing, and RTM setup and execution (shown

in Figure 1) are automated such that the 3D/1D-RT calculations can be performed for any region
at any date and time using satellite or aircraft data or other data resources such as LES. A detailed
code walk-through of App. 1 and 2 is provided in Appendix A2. Since EaR$^3$T is developed as an
educational and research 3D-RT tool collection by students, it is a living code base, intended to be
updated over time. The master code modules for the five applications as listed in Figure 1 are
included in the EaR$^3$T package under the `examples` directory. In the current release (v0.1.1),
only a limited documentation for the installation and usage, including example code for EaR$^3$T, is
provided. More effort will be dedicated for documentation in the near-future.

In the following sections, we discuss results obtained from EaR$^3$T, starting with those from

`examples/01_oco2_rad-sim.py` and `examples/02_modis_rad-sim.py` (section
4), `examples/03_spns_flux-sim.py` (section 5), and concluding with
`examples/04_cam_nadir_rad-sim.py` (section 6). The usage of the EaR$^3$T package
including the technical input and output parameters and code walk-through is provided in
Appendix A.

**438    4. EaR$^3$T as a 3D Satellite Radiance Simulator**

This section demonstrates the automated 3D radiance simulation for satellite instruments

by EaR$^3$T for OCO-2 and MODIS measured radiance based on publicly available MODIS retrieval
products. The OCO-2 application is an example of radiance consistency between two distinct
satellite instruments where the measurements of one (here, OCO-2) are compared with the
simulations based on data products from the other (here, MODIS). The MODIS application, on
the other hand, is an example of radiance self-consistency. We will show how inconsistencies can
be used for detecting cloud and surface property retrieval biases.

**446    4.1 OCO-2 (App. 1)**

The OCO-2 radiance measurements at 768.52 nm for our sample scene in the context of

MODIS imagery were shown in Figure 2. For that track segment, Figure 5a shows the simulated
radiance along with the measurements as a function of latitude. The radiance was averaged over
every 0.01° latitude window from 37° N to 39° N (the standard deviation within the bin indicated
by the shaded color). In clear-sky regions (e.g., around 38.2º N), the 3D simulations (red) are
systematically higher than the measurements (black), even though the footprint-level OCO-2
surface albedo retrieval was used to replace and scale the MCD43 surface albedo field as described
in section 2.2.2 (Figure 4). This is probably because, unlike the MCD43 algorithm which relies on
multiple overpasses and multiple-days for cloud-clearing, the OCO-2 retrieval is done for any clear
footprint. Clouds in the vicinity lead to enhanced diffuse illumination that is erroneously attributed
to the surface albedo itself. The EaR$^3$T IPA calculations of the clear-sky pixels (blue) essentially
reverse the 3D effect and therefore match the observations better. The 3D calculations enhance the
reflectance through the very same 3D cloud effects that led to the enhanced surface illumination
in the first place. It is possible to correct this effect by down-scaling the surface albedo according
to the ratio between clear-sky 3D and IPA calculations, but this process is currently not automated.

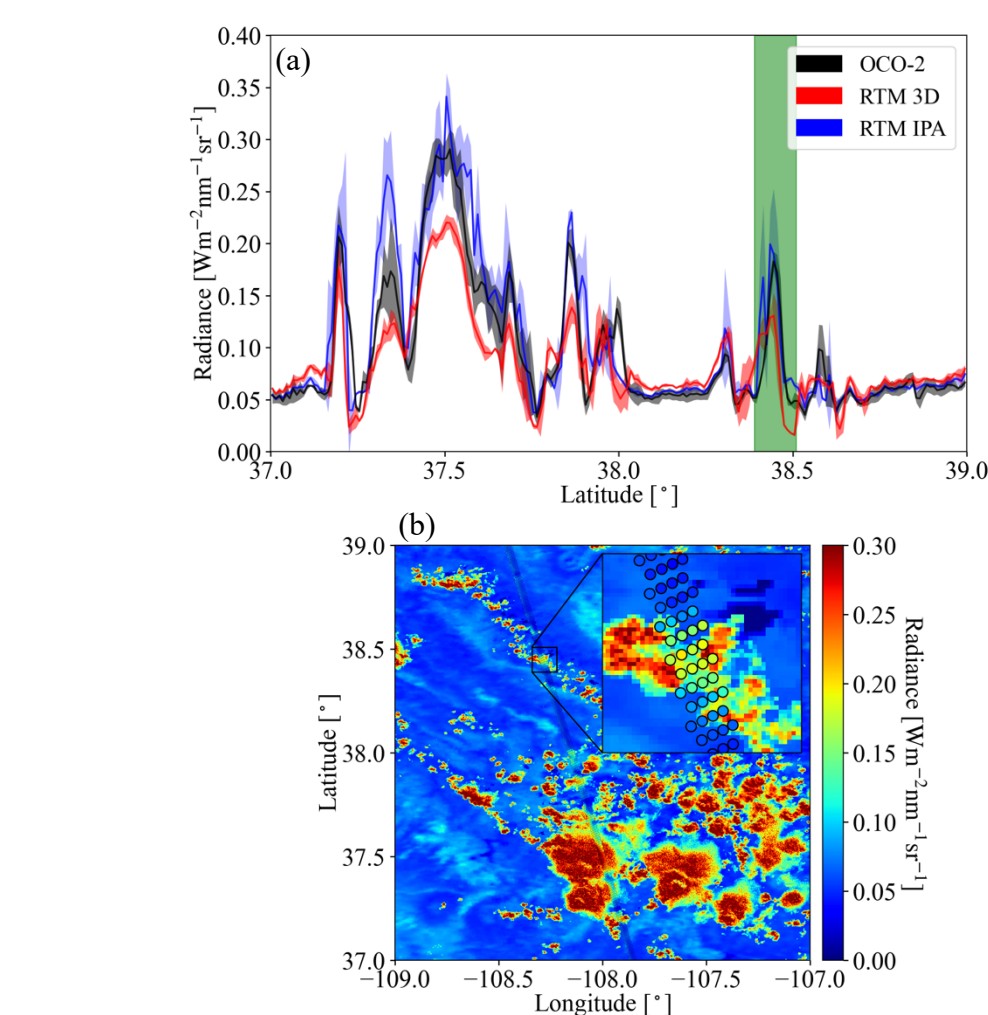



**Figure 5**. **(a)** Latitudinally averaged (0.01° spacing) radiance calculations from EaR$^3$T (red: 3D, blue: IPA) and OCO-
2 measured radiance at 768.52 nm (black) The green shaded area indicates the inset shown in (b). **(b)** The

same as Figure 2 except OCO-2 measured radiance overlaid on IPA radiance simulations at 768.52 nm. The

solar zenith angle (SZA) for the radiance simulation case is 34.3°.


In the cloudy locations (radiance value greater than ~0.05), the IPA calculations match the

OCO-2 observations on a footprint-by-footprint level (see Figure 5b), demonstrating that wind and
parallax corrections were performed successfully. Of course, there is not always a perfect
agreement because of morphological changes in the cloud field over the course of six minutes. It
is, however, apparent that the 3D calculations agree to a much lesser extent with the observations
than the IPA calculations. Just like the mismatch for the clear-sky pixels indicates a bias in the
input surface albedo, the bias here means that the input cloud properties (most importantly COT)
are inaccurate. For most of the reflectance peaks, the 3D simulations are too low, which means
that the input COT is biased low. This is due to 3D cloud effects on the MODIS-based cloud
retrieval. Since they are done with IPA, any net horizontal photon transport is not considered,
which leads to an apparent surface brightening as noted above, at the expense of the cloud
brightness. As a result, the COT from darker clouds is significantly underestimated. This
commonly known problem (Barker and Liu, 1995), with several aspects discussed in the
subsequent EaR[3]T applications, can be identified by radiance consistency checks such as the one
shown in Figure 5, and mitigated by novel types of cloud retrievals that do take horizontal photon
transport into account (section 6).

**4.2 MODIS (App. 2)**

To go beyond the OCO-2 track and understand the bias between simulated and observed

radiances from a domain perspective, we now consider the radiance simulations for the MODIS
650 nm channel. The setup is exactly the same as for the OCO-2 simulations, except that 1) the
viewing zenith angle is set to the average viewing zenith angle of MODIS within the shown domain
(instead of OCO-2), and 2) the surface albedo (or WSA) from MCD43 is used directly, this time
from the 650 nm channel without rescaling. Figure 6a shows the MODIS measured radiance field,
while Figure 6b shows the EaR[3]T 3D simulations. Visually, the clouds from the EaR[3]T simulation
are generally darker than the observed clouds, which is in line with our aforementioned explanation
of net horizontal photon transport. They are also blurrier because radiative smoothing (Marshak et
al., 1995) propagates into the retrieved COT fields, which are subsequently used as input to EaR[3]T.
The IPA RT calculations agree with the observations for clouds (see Figure A4a in Appendix C2),
which is expected as the IPA calculations and retrievals go through the same RT process, and the
darkening and smoothing effects (referred to as 3D effects) are due to horizontal photon transport.
To look at the 3D effects more quantitatively, Figure 7 shows a heatmap plot of simulated radiance
versus observed radiance. It shows that the radiance for cloud-covered pixels (labeled "cloudy")
from EaR$^3$T are mostly low-biased while good agreement between simulations and observations
was achieved for clear-sky radiance (labeled "clear-sky"). The good agreement over clear-sky
regions is expected. As mentioned above, we use MCD43 as surface albedo input, which in
contrast to the OCO-2 surface albedo product is appropriately cloud-screened and therefore does
not have a reflectance high bias. There is, of course, a reflectance enhancement in the vicinity of
clouds, but that is captured by the EaR$^3$T calculations. The fact that the calculations agree with the
observations even for clear-sky pixels in the vicinity of clouds, shows that the concept of radiance
consistency works to ensure correct satellite retrievals even in the presence of clouds. It also
corroborates our observation from section 4.1 that $COT_{IPA}$ is low biased. Since the MODIS
reflectance is *not* self-consistent with respect to 3D RT calculations using $COT_{IPA}$ as shown for
the *cloudy* pixels in Figure 7, we can identify a bias in the cloud properties even without knowing
the ground truth of COT. On the other hand, successful closure in radiance (self-consistency)
would provide an indication that the input fields including COT are accurate, although it is
certainly a weaker metric than direct verification of the retrievals through aircraft-satellite retrieval
validation using observations from in-situ instruments.

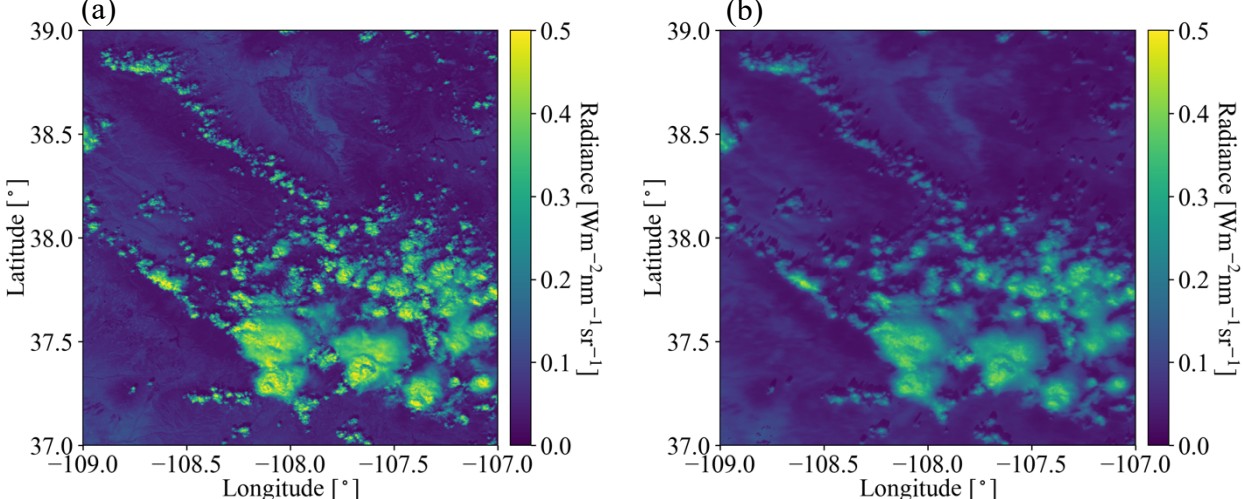


**Figure 6**. **(a)** MODIS measured radiance in channel 1 (650 nm). **(b)** Simulated 3D radiance at 650 nm from EaR³T.

The solar zenith angle for the radiance simulation case is 34.94°.



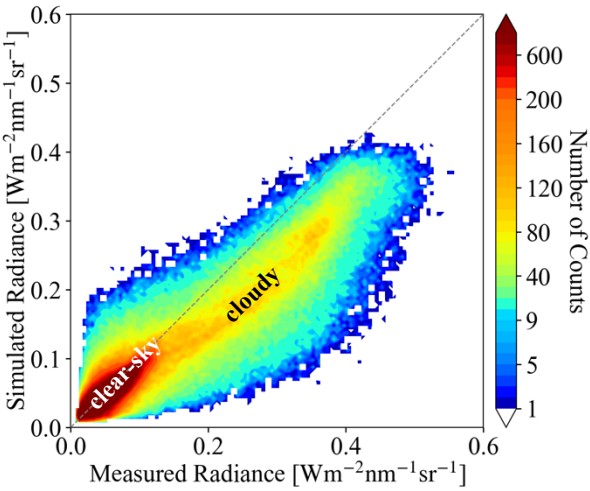


**Figure 7**. Heatmap plot of EaR³T simulated 3D radiance vs. MODIS measured radiance at 650 nm.

Summarizing the two satellite radiance simulator applications, one can say that EaR³T

enables a radiance consistency check for inhomogeneous cloud scenes. We demonstrated that a
lack of simulation-observation consistency (MODIS versus OCO-2) and self-consistency (MODIS
versus MODIS) can be traced back to biased surface albedo or cloud fields in the simulator input.
This can become a diagnostic tool for the quality of retrieval products from future or current
missions, even when the ground truth is not known. Although not shown, the errors in the
simulated radiance associated with the fixed-SZA assumption (domain average) are negligible.
However, the vertical extent of the clouds affects the simulated radiance – the larger the vertical
extent, the larger the 3D effects (more horizontal photon transport). Since we make the assumption
of 1) a cloud geometric thickness of 1 km for clouds with CTH less than 4 km, and 2) cloud base
height of 3 km for clouds with CTH greater than 4km, the simulated radiance at the satellite sensor
level is valid for that proxy cloud only. For clouds that are geometrically thicker than the assumed
cloud geometrical thickness, the simulated radiance would be even lower due to enhanced
horizontal photon transport. Either way, the comparison with the actual radiance measurements
will reveal a lack of closure. Additionally, although the clouds introduce the lion's share of the 3D
bias that is identified by the radiance consistency check, additional discrepancies can be introduced
in different ways. For example, the topography (mountainous region in Colorado) is not considered
by MCARaTS (it is considered by MYSTIC, but this solver has not been implemented yet).
For the reference of simulation running time: The MODIS simulation (domain size of
[Nx=846, Ny=846]) took about 15 minutes on a Linux workstation with 8 CPUs for three 3D RT
runs with $10^8$ photons. With a slightly modified setup and parallelization, the automation can be
easily applied for entire satellite orbits, although more research is required to optimize the
computation speed depending on the desired output accuracy.

**5. EaR³T as 3D Aircraft Irradiance Simulator (App. 3)**
In contrast to the previous applications that focused on satellite remote sensing, we will
now be applying EaR³T to quantify 3D cloud retrieval biases through direct, systematic validation
of imagery-derived *irradiances* against aircraft measurements, instead of using the indirect path
of radiance consistency in section 4. Previous studies (e.g., Schmidt et al., 2007; Kindel et al.,
2010) conducted radiative closure between remote sensing derived and measured irradiance using
isolated flight legs as case studies. Here, with the efficiency afforded by the automated nature of
EaR³T, we are able to conduct radiative closure of irradiance through a statistical approach that
employs campaign-scale amounts of measurement data. Specifically, we used EaR³T to perform
large-scale downwelling irradiance simulations at 745 nm based on geostationary cloud retrievals
from AHI for the CAMP²Ex campaign, and directly compare these simulations to the SPN-S
measured irradiances onboard the P-3 aircraft. This is done for all below-cloud legs from the entire
campaign with the aim to assess the degree to which satellite-derived near-surface irradiances
reproduce the true conditions below clouds.
The irradiance simulation process is similar to the previously described radiance simulation
in section 4, with only a few modifications. First, we used cloud optical properties from the AHI
cloud product (COT, CER and CTH) as direct inputs into EaR³T. Secondly, we used a constant
ocean surface albedo value of 0.03. Such simplification in surface albedo is made under the
assumption that 1) the ocean surface is calm with no whitecaps, and that 2) the Lambertian BRDF
is sufficient (instead of directionally dependent BRDF) to represent surface albedo for the
irradiance calculation. Since the ocean surface albedo can greatly differ from 0.03 when the Sun
is extremely low (Li et al., 2006), we excluded data under low-Sun conditions where the SZA is
greater than 45°. Lastly, since EaR³T can only perform 3D simulations for a domain at a single
specified solar geometry, we divided each CAMP[2]Ex research flight into small flight track
segments where each segment contains 6 minutes of flight time. The size and shape of the flight
track segments can vary significantly due to the aircraft maneuvers, aircraft direction, aircraft
speed, etc. For each flight track segment, EaR$^3$T performs irradiance simulations for a domain that
extends half a degree at an averaged solar zenith angle. In contrast to the radiance simulation output,
which is two-dimensional at a specified altitude and sensor geometry, the irradiance simulation
output is three dimensional. In addition to x (longitude) and y (latitude) vectors, it has a vertical
dimension along z (altitude). From the simulated three-dimensional irradiance field, the irradiance
for the flight track segment is linearly interpolated to the x-y-z location (longitude, latitude, and
altitude) of the aircraft. EaR$^3$T automatically sub-divides the flight track into tiles encompassing
track segments, and extracts the necessary information from the aircraft navigational data. Based
on the aircraft time and position, EaR$^3$T downloads the AHI cloud product that is closest in time
and space to the domain containing the flight track segment.

Figure 8 shows the simulated irradiance for a sample flight track below clouds on 20

September, 2019. Figure 8a shows the flight track overlaid on AHI imagery. Figure 8b shows 3D
(in red) and IPA (in blue) downwelling irradiance simulations for the highlighted flight track in
Figure 8a, as well as measurements by the SPN-S (in black). Since the 3D and IPA simulations
are performed separately at discrete solar and sensor geometries for each flight track segment based
on potentially changing cloud fields from one geostationary satellite image to the next,
discontinuities in the calculations (indicated by gray dashed lines) are expected. The diffuse
irradiance (downwelling and upwelling) can also be simulated and compared with radiometer
measurements (not shown here). Since the irradiance was simulated/measured below clouds, high
values of downwelling irradiance indicate thin-cloud or cloud-free regions while low values of
downwelling irradiance indicate thick-cloud regions. The simulations successfully captured this
general behavior – clouds thickened from west to east until around 121.25° E, and thinned
eastwards. However, the fine-scale variabilities in irradiance were not captured by the simulations
due to the coarse resolution of COT in the AHI cloud product (3-5 km). Additionally, the
simulations also missed the clear-sky regions in the very east and west of the flight track as
indicated by high downwelling irradiance values measured by SPN-S. This is probably also due to
the coarse resolution of the AHI COT product where small cloud gaps are not represented. Large
discrepancies between simulations and observations occur in the mid-section of the flight track
where clouds are present (e.g., longitude range from 121.15° to 121.3°). Although the 3D
calculations differ somewhat from the IPA results, they are both biased high, likely because the
input COT (the IPA-retrieved AHI product) is biased low. This bias is caused by the same
mechanism that was discussed earlier in the MODIS examples (section 4.2). This begs the question
whether this is true for the entire field mission. To answer the question, we performed a *systematic*
comparison of the cloud transmittance for *all* available below-cloud flight tracks from CAMP[2]Ex,
using EaR[3]T's automated processing pipeline. The output of this pipeline is visualized in time-
synchronized flight videos (Chen et al., 2022), which show the simulations and observations along
all flight legs point by point. These videos give a glimpse of the general cloud environment during
the field campaign from the geostationary satellite perspective.


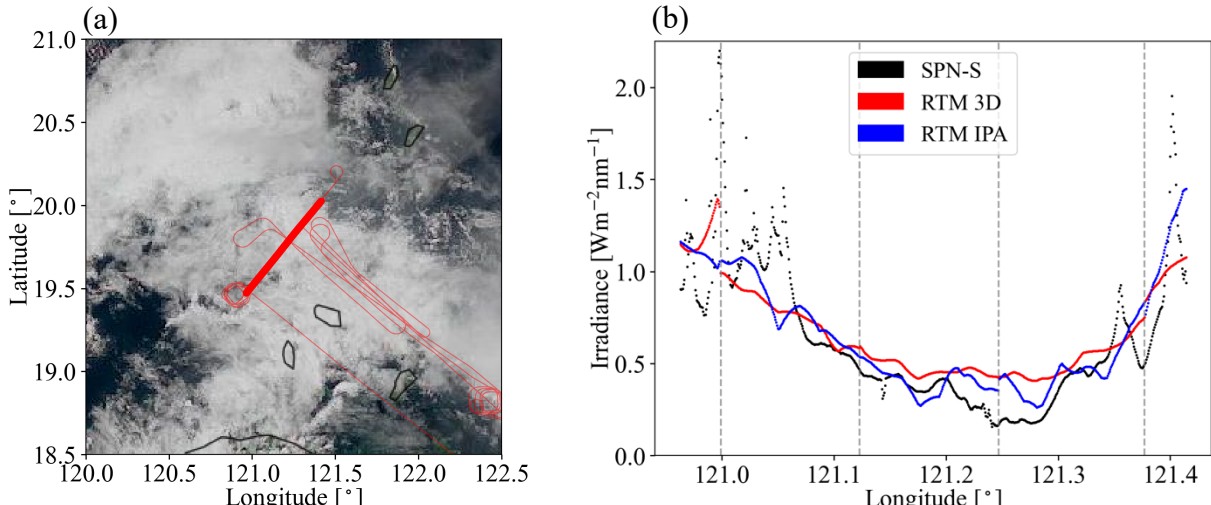

**Figure 8**. **(a)** Flight track overlay HIMAWARI AHI RGB imagery over the Philippine Sea on 20 September, 2019.

The thin line shows the entire flight track within the domain. The thick line highlights the specific leg

analyzed in (b). **(b)** Measured downwelling irradiance from SPN-S at 745 nm and calculated 3D and IPA

irradiance from EaR[3]T for the highlighted flight track in (a).


For this comparison, we use transmittance instead of irradiance. The transmittance is
calculated by dividing the downwelling irradiance below clouds ($F_\downarrow^{bottom}$) by the downwelling
irradiance at the top of the atmosphere extracted from the Kurucz solar spectra ($F_\downarrow^{TOA}$; Kurucz,
1992) at incident solar zenith angle (SZA), where
$$Transmittance = \frac{F_\downarrow^{bottom}}{F_\downarrow^{TOA} \cdot \cos(SZA)}$$
Thus the transmittance has less diurnal dependence than the irradiance. Figure 9 shows the
histograms of the simulated and measured cloud transmittance from all below-cloud legs. The
average values are indicated by dashed lines. Although the averaged values of IPA and 3D
transmittance are close, their distributions are different. Only the 3D calculations and the measured
transmittance reach values beyond 1. This occurs in clear-sky regions in the vicinity of clouds that
receive photons scattered by the clouds as previously discussed for the OCO-2 application.

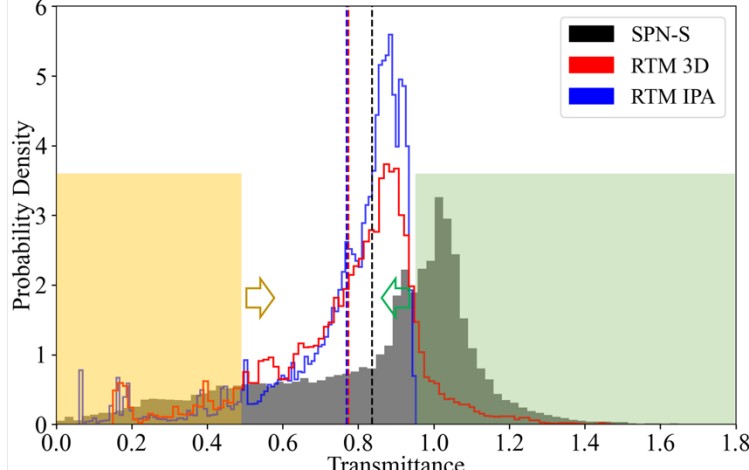


**Figure 9**. Histogram of measured transmittance from SPN-S at 745 nm (dark gray filled) and calculated 3D (red solid
line) and IPA (blue solid line) transmittance from EaR[3]T for all the below-cloud flight tracks during
CAMP[2]Ex in 2019. The mean values are indicated by dashed lines. The yellow (green) shaded area
represents the relatively low (high) transmittance region where the probability density of the observed
transmittance (dark gray filled) is greater than the calculations.

Both the distribution and the mean value of the simulations are different from the
observations – the simulation histograms peak at around 0.9 while the observation histogram peaks
at around 1. The histograms indicate that the RT simulations miss most of the clear-sky conditions
because of the coarse resolution of the AHI cloud product. If clouds underfill a pixel, AHI
interprets the pixel as cloudy in most cases. This leads to an underestimation of clear-sky regions
since cumulus and high cirrus were ubiquitous during CAMP[2]Ex. The area on the left (highlighted
in yellow) has low cloud transmittance associated with thick clouds. In this range, the histograms
of the calculations are generally below the observations, and the PDF of the calculations is offset
to the right (indicated by the yellow arrow). This means that the transmittance is overestimated by
both IPA and 3D RT, and thus that the COT of thick clouds is underestimated, consistent with
what we found before (Figure 8b). The high-biased transmittance below-cloud is also consistent
with the findings of low-biased reflectance (App. 1 and 2), both indicating COT of the optically
thick clouds are low-biased. The high-transmittance end (highlighted in green) is associated with
clear-sky and thin clouds. Here, the peak of the PDF is shifted to the left (green arrow), and the
calculations are biased low. This is caused by a combination of 1) the overestimation in COT of
thin clouds due a 3D bias in the AHI IPA retrieval, 2) the aforementioned resolution effect that
underestimates the occurrence of clear-sky regions (or overestimation in cloud fraction), and 3)
net horizontal photon transport from clouds into clear-sky pixels. Overall, the calculations
underestimate the true transmittance by 10%. This might seem to contradict Figure 7, where the
calculated reflected radiance was biased low due to the *underestimation* of COT in the heritage
retrievals, which would correspond to an *overestimation* of the radiation transmitted by clouds.
This effect is indeed apparent in the yellow-shaded area of Figure 9 (high COTs), but the means
(dashed lines) show exactly the opposite. To understand that, one has to consider that the histogram
depicts all-sky conditions, which include both cloudy and clear pixels. In this case, the direction
of the overall (all-sky) bias follows the direction of the thin-cloud/clear bias, rather than the
direction of the thick cloud bias. For different study regions of the globe with different cloud
fractions, cloud size distributions, and possibly different imager resolutions, the direction and
magnitude of the bias might be very different.

Summarizing, this application demonstrates that the EaR$^3$T's automation feature allows

systematic simulation-to-observation comparisons. If aircraft observations are available, then
closure between satellite-derived irradiance and suborbital measurements is a more powerful
verification of satellite cloud retrieval products than the radiance consistency from the earlier
stand-alone satellite applications. Even more powerful is the new approach to process the data
from an entire field mission for assessing the quality of cloud products in a region of interest (in
this case, the CAMP$^2$Ex area of operation).

**6. EaR$^3$T for Mitigating 3D Cloud Retrieval Biases (App. 4)**

In this section, we will use high-resolution imagery from a radiometrically calibrated

all-sky camera flown during the CAMP$^2$Ex to isolate the 3D bias (sometimes referred to as IPA
bias) and explore its mitigation with a newly developed CNN cloud retrieval framework (Nataraja
et al., 2022). The CNN, unlike IPA, takes pixel-to-pixel net horizontal photon transport into
account. It exploits the spatial context of pixels in cloud radiance imagery, and extracts a higher-
dimensional, multi-scale representation of the radiance to retrieve COT fields as the output. It does
so by learning on "training data", which in this case was input radiance and COT pairs synthetically
generated by EaR$^3$T using LES data from the Sulu Sea. The best CNN model, trained on different
coarsened resolutions of the data pairs, is included within the EaR$^3$T repository. For App. 4, this
CNN is applied to real imagery data for the first time, which in our case are near-nadir observations
by the all-sky camera (section 2.2.5) that flew in CAMP$^2$Ex.

The CNN model was trained at a single (fixed) sun-sensor geometry (solar zenith angle,

SZA=29.2°; solar azimuth angle, SAA=323.8°, viewing zenith angle, VZA=0º), at a spatial
resolution of 100 m. We therefore chose a camera scene with a matching SZA (28.9°), and rotated
the radiance imagery to match SAA=323.8°, and subsequently gridded the 8-12 m native
resolution camera data to 100 m. Figure 10a shows the RGB imagery captured by the all-sky
camera over the Philippine Sea at 02:10:06 UTC on 5 October 2019. The Sun is located at the
southeast (as indicated by the yellow arrow) and can be easily identified from the sun glint. Note
that this image has not yet been geolocated; it is depicted as acquired in the aircraft reference frame.
Figure 10b shows the rotated scene of the red channel radiance for the region encircled in yellow
in Figure 10a. The sun (as indicated by the yellow arrow) is now at SAA=323.8°. The selected
study region is indicated by the red rectangle in Figure 10b (6.4x6.4 km$^2$), where the raw radiance
of the camera is gridded at 100 m resolution to match the spatial resolution of the training dataset
of the CNN.


(a)

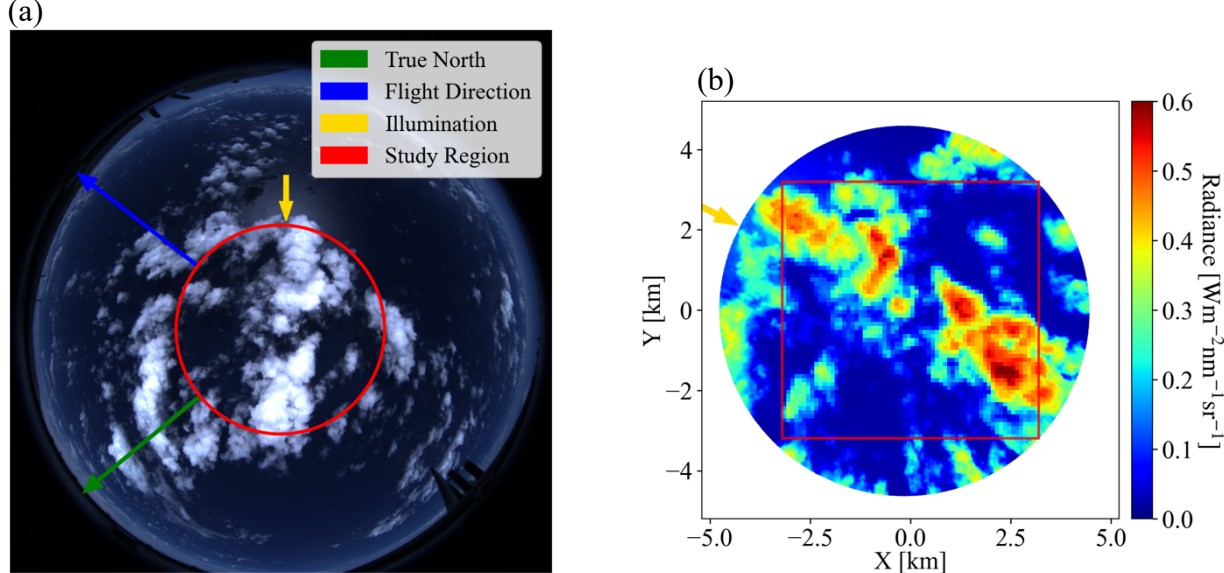

Figure 10. **(a)** RGB imagery of nadir-viewing all-sky camera deployed during CAMP²Ex for a cloud scene centered at [123.392°E, 15.2744°N] over the Philippine Sea at 02:10:06 UTC on 5 October, 2019. The arrows indicate the true north (green), flight direction (blue), and illumination (where the sunlight comes from, yellow). **(b)** Red channel radiance measured by the camera for the circular area indicated by the red circle in (a). Red squared region shows gridded radiance with a pixel size of 64x64 and spatial resolution of 100 m.

From the radiance field, we used both the traditional IPA (based on the IPA reflectance-to-COT mapping) and the new CNN to retrieve COT fields. Figure 11 shows the $COT_{IPA}$ and $COT_{CNN}$ fields, which are visually quite different. For relatively thin clouds (e.g., at around {2, 1.8}), the CNN tends to retrieve larger COT values than $COT_{IPA}$. Also, it returns more spatial structure than the IPA (e.g., around {2,-1}). To assess how either retrieval performs, we now apply the radiance self-consistency approach introduced with MODIS data in section 4.2. Using both the IPA and the CNN retrieval as input, we had EaR³T calculate the (synthetic) radiance that the camera should have observed if the retrieval were accurate. The clouds are assumed to be located at 1-2 km. Such an assumption is inferred from low-level aircraft observations of clouds on the same day. These radiance fields are shown in Figure 12a and 12b, and can be compared to Figure 12c. Seven edge pixels have been removed from the original domain because the CNN performs poorly at edge pixels, and because the 3D calculations use periodic boundary conditions.

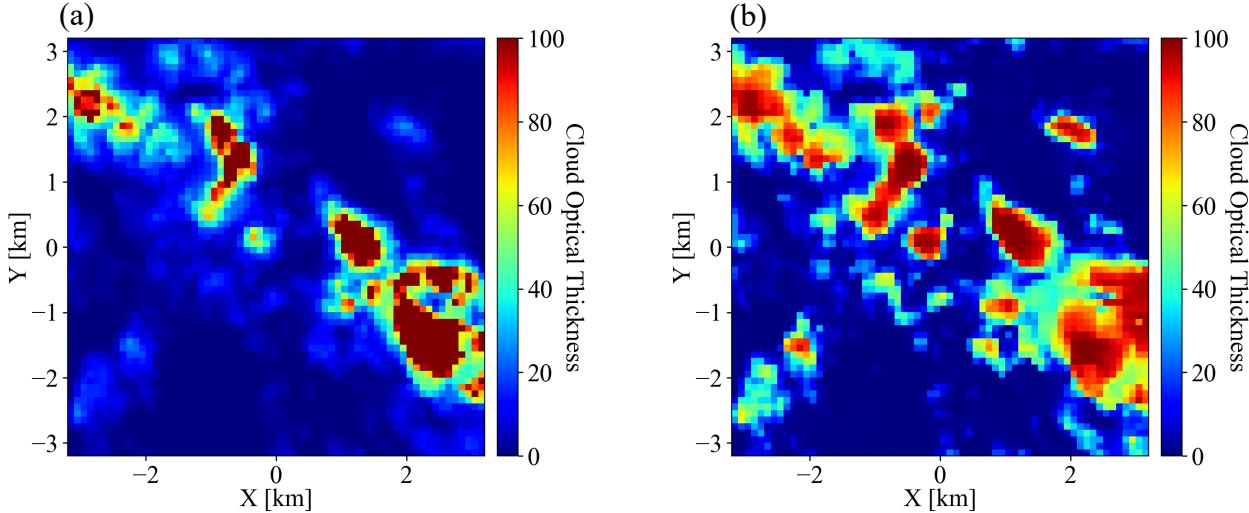


**Figure 11**. Cloud optical thickness for the gridded radiance in Figure 10b **(a)** estimated by IPA method and **(b)**

predicted by CNN.





**Figure 12**. 3D radiance calculations from EaR³T at 600 nm based on cloud optical thickness field **(a)** estimated by
IPA, and **(b)** predicted by the CNN. The radiance measured by the all-sky camera (the same as Figure
10b) is provided in the same format at **(c)** for comparison. The calculations were originally performed
for the 64x64 domain. Then 7 pixels along each side of the domain (contoured in gray) were excluded,
which resulted in a 50x50 domain.



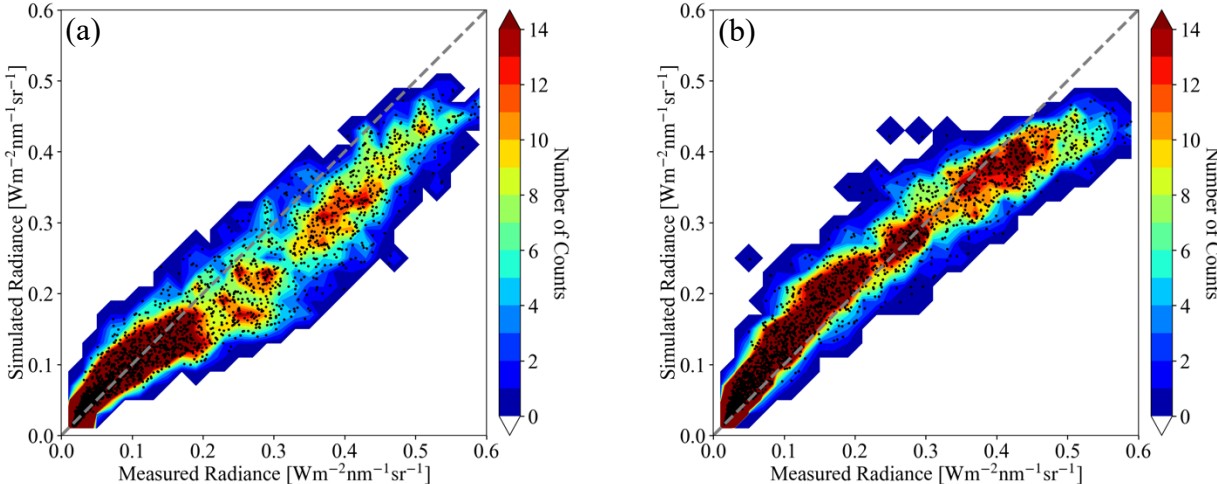


**Figure 13**. Scatter plot overlays 2D histogram of 3D radiance calculations at 600 nm based on cloud optical thickness
**(a)** estimated by IPA and **(b)** predicted by the CNN vs. measured red channel radiance from all-sky camera.


As evident from the brightest pixels in Figures 12b and 12c, the radiances simulated on the

basis of the COT$_{CNN}$ input are markedly lower than actually observed by the camera. This is
because the CNN was trained on a LES dataset with limited COT range that excluded the largest
COT that occurred in practice. This means that the observational data went beyond the original
training envelope of the CNN, which highlights the importance of choosing the CNN training data
carefully for a given region. In Figure 13, the simulations are directly compared with the original
observations, confirming that indeed the CNN-generated data are below the observations on the
high radiance end. Otherwise, the CNN-generated radiances agree with the observations. In
contrast, the IPA-generated data are high biased for the optically very thin clouds (radiance below
0.1) and systematically low-biased for the thick clouds (radiance above 0.2) when comparing with
the observations, over the dynamic range of the COT, which is indicative of the 3D retrieval bias
that we discussed earlier. A small high bias occurs in the COT$_{CNN}$ based radiance simulations for
the optically thin clouds (radiance value below 0.2). This probably because the CNN training as
described by Nataraja et al. (2022) is 1) based on a surface albedo of 0 and 2) aerosol-free
atmospheric environment (also aerosol-free setup for radiance simulations in Figure 13), where in
reality the ocean is slightly brighter and atmosphere is mixed with aerosols. Here again, the
radiance self-consistency approach proves useful despite the absence of ground truth data for the
COT. This is valuable because in reality satellite remote sensing does not have the ground truth of
COT, whereas radiance measurements are always available. For the CNN, the self-consistency of
the radiance is remarkable for most of the clouds (radiance smaller than 0.4), which encompass
86.8% of the total number of image pixels.
Finally, we use EaR$^3$T to propagate the 3D cloud retrieval bias into the associated bias in
estimating the cloud radiative effect from passive imagery retrievals, which means that we are
returning from a remote sensing to an energy perspective (irradiance) at the end of the paper. The
calculated cloud radiative effects (CRE) of both below-clouds (at the surface) and above-clouds
(at 2.5 km) are shown in Figure 14a and 14b. The most important histograms are those from 3D
irradiance calculations based on the CNN retrievals (gray solid line), as this combination would
be used in a next-generation framework for deriving CRE from passive remote sensing, and the
other would be IPA irradiance calculations based on the IPA retrieval (red solid line), as done in
the traditional (heritage) approach. The dashed lines are the other combinations. The mean values
(red vs. gray) indicate that in our case the traditional approach would lead to a high bias of more
than to 28% both at the surface and 20% above clouds due to low-biased $COT_{IPA}$ (consistent with
findings of low-biased $COT_{IPA}$-derived reflectance from App. 1&2 and high-biased $COT_{IPA}$-
derived transmittance from App. 3). Here again, 3D biases do not cancel each other out in the
domain average. If the CNN had better fidelity even for optically thick clouds, the real bias in CRE
would be even larger. A minor, but interesting finding is that regardless of which COT retrieval is
used, the mean CRE is similar for IPA and 3D irradiance calculations (e.g., $\overline{CRE_{IPA}(COT_{CNN})} \approx$
$\overline{CRE_{3D}(COT_{CNN})}$, blue vertical dashed line locates near to gray vertical solid line), even though
the PDFs are different. By far the largest impact on accuracy comes from the retrieval technique,
not from the subsequent CRE calculations. Here again, the self-consistency check turns out as a
powerful metric to assess retrieval accuracy. Of course, we only used a single case in this part of
the paper. For future evaluation of the CNN versus the IPA, one would need to process larger
quantities of data in an automated fashion as done in the first part of the paper. This is beyond the
scope of this introductory paper, and will be included in future releases of EaR$^3$T and the CNN.

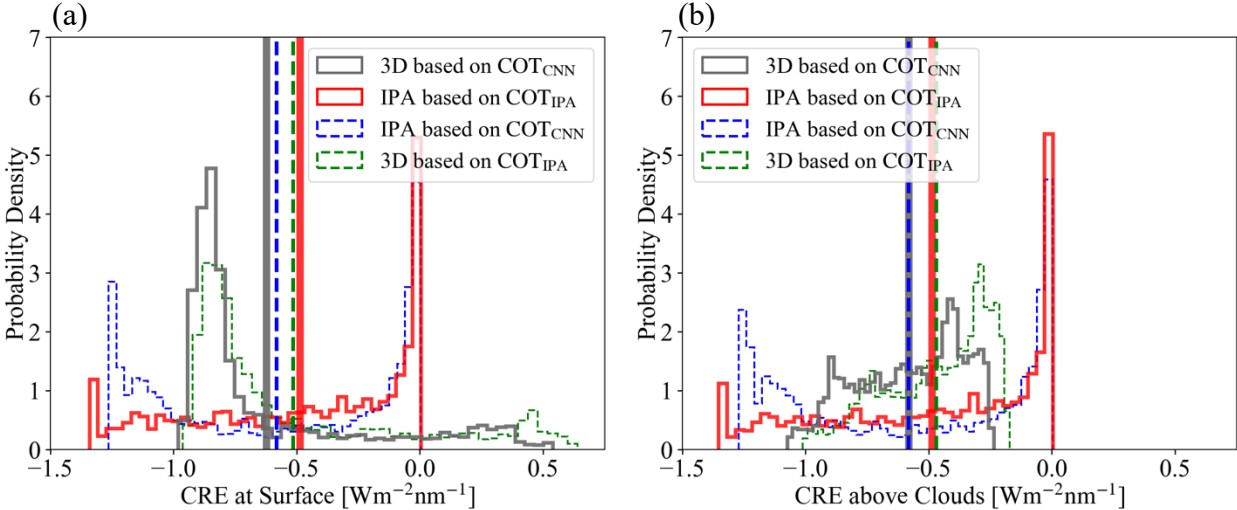


**Figure 14**. Histograms of cloud radiative effects derived from 1) 3D irradiance calculations based on COT$_{CNN}$ (solid

gray), 2) IPA irradiance calculations based on COT$_{IPA}$ (solid red), 3) IPA irradiance calculations based on

COT$_{CNN}$ (dashed blue), and 4) 3D irradiance calculations based on COT$_{IPA}$ (dashed green) both **(a)** at the

surface and **(b)** above the clouds. The mean values are indicated by vertical lines.


**7. Summary and Conclusion**

In this paper, we introduced EaR$^3$T, a toolbox that provides high-level interfaces to

automate and facilitate 1D- and 3D-RT calculations. We presented applications that used EaR$^3$T
to:
a) build a processing pipeline that can automatically simulate 3D radiance fields for satellite

instruments (currently OCO-2 and MODIS) from publicly available satellite surface and

cloud products at any given time over any specific region;

b) build a processing pipeline that can automatically simulate irradiance along all flight legs

of aircraft missions, based on geostationary cloud products;

c) simulate radiance and irradiance for high-resolution COT fields retrieved from an airborne

camera, using both a traditional 1D-RT (IPA) approach, and a newly developed 3D-RT

(CNN) approach that considers the spatial context of a pixel.

Unlike other satellite simulators that employ 1D-RT, EaR$^3$T is capable of performing the radiance
and irradiance calculations in 3D-RT mode. Optionally, it can be turned off to link back to
traditional 1D-RT codes, and to calculate 3D perturbations by considering the changes of 3D-RT
fields relative to the 1D-RT baseline.
With the processing pipeline under a) (App. 1 and App. 2, section 4), we prototyped a
3D-RT powered radiance loop (we call it "radiance self-consistency") that is envisioned for
upcoming satellite missions such as EarthCARE and AOS. Retrieved cloud fields (in our case,
from MODIS and from an airborne camera) are fed back into a 3D-RT simulation engine to
calculate at-sensor radiances, which are then compared with the original measurements. Beyond
currently included sensors, others can be added easily, taking advantage of the modular design of
EaR$^3$T. This radiance closure loop facilitates the evaluation of passive imagery products,
especially under spatially inhomogeneous cloud conditions. The automation of EaR$^3$T permits
calculations at any time and over any given region, and statistics can be built by looping over entire
orbits as necessary. The concept of radiance self-consistency could be valuable even for existing
imagery datasets because it allows the automated quantification of 3D-RT biases even without
ground truth such as airborne irradiance from suborbital activities. Also, it can be easily extended
to spectral or multi-angle observations as available from MODIS and MISR (Multi-Angle Imaging
Spectroradiometer), and thus providing more powerful constraints to the remote sensing products.
In the future it should be possible to include a 3D-RT pipeline such as EaR$^3$T into operational
processing of satellite derived data products.
Benefitting from the automation of EaR$^3$T in b) (App. 3, section 5), we performed 3D-RT
irradiance calculations for the entire CAMP$^2$Ex field campaign, moving well beyond radiation
closure case studies, and instead systematically evaluating satellite-derived radiation fields with
aircraft data for an entire region. From the comparison based on all below-cloud flight tracks
during the entire campaign, we found that the satellite-derived cloud transmittance was biased low
by 10% compared to the observations when relying on the heritage satellite cloud product.
From the statistical results of the CAMP$^2$Ex irradiance closure in b), we concluded that the
bias between satellite-derived irradiances and the ground truth from aircraft measurements was
due to a combination of the coarse spatial resolution of the geostationary imagery products and
3D-RT effects. To minimize the coarse-resolution part of the bias and thus to isolate the 3D-RT
bias, we used high-resolution airborne camera imagery in c) (App. 4, section 6), and found that
even with increased imager resolution, biases persisted. The at-sensor radiance derived from
$COT_{IPA}$ was inconsistent with the original measurements. For cloudy pixels, the calculated
radiance was well below the observations, confirming an overall low bias in $COT_{IPA}$. This low bias
could be largely mitigated with the context-aware CNN developed separately in Nataraja et al.
(2022) and included in EaR$^3$T. Of course, this novel technique has limitations. For example, the
camera reflectance data went beyond the CNN training envelope, which would need to be extended
to larger COT in the future. In addition, the CNN only reproduces two-dimensional clouds fields
and does not provide access to the vertical dimension, which will be the next frontier to tackle.
Still, the greatly improved radiance consistency from COT$_{IPA}$ to COT$_{CNN}$ indicates that the EaR$^3$T-
LES-CNN approach shows great promise for the mitigation of 3D-RT biases associated with
heritage cloud retrievals. We also discovered that for this particular case, the CRE calculated from
traditional 1D cloud products can introduce a warm bias of at least 28% at the surface and 20%
above clouds.
EaR$^3$T has proven to be capable of facilitating 3D-RT calculations for both remote sensing
and radiative energy studies. Beyond the applications described in this paper, EaR$^3$T has already
been extensively used by a series of on-going research projects such as producing massive 3D-RT
calculations as training data for a new generation of CNN models (Nataraja et al., 2022), evaluating
3D cloud radiative effects associated with aerosols (Gristey et al., 2022), creating flight track and
satellite track simulations for mission planning etc. More importantly, the strategies provided in
this paper put novel machine learning algorithms on a physical footing, opening the door for the
mitigation of complexity-induced biases in the near-future. More development effort will be
invested into EaR$^3$T in the future, with the goals of minimizing the barriers to using 3D-RT
calculations, and to promote 3D cloud studies. EaR$^3$T will continue to be an educational tool driven
by graduate students. In the future, we plan to add support for additional publicly available 3D RT
solvers, e.g., SHDOM (Spherical Harmonic Discrete Ordinate Method, Evans, 1998; Pincus and
Evans, 2009), as well as built-in support for HITRAN and associated correlated-k methods
(currently, we are implementing such an approach for the longwave wavelength range). From a
research perspective, we anticipate that EaR$^3$T will enable the systematic quantification and
mitigation of 3D-RT biases of imagery-derived cloud-aerosol radiative effects, and may be the
starting point for operational use of 3D-RT for future satellite missions.

 **Appendix A**

**A1 - Technical Input and Output Parameters of EaR³T**
EaR³T provides various functions that can be combined to tailored pipelines for automatic
3D radiative transfer (3D-RT) calculations as described in this paper (App. 1 – 5), as well as for
complex research projects beyond. Since EaR³T is written in Python, the modules and functions
can be integrated into existing functions developed by the users themselves. Parallelization is
enabled in EaR³T by default through multi-processing to accelerate computations. If multiple
CPUs are available, EaR³T will distribute jobs for the 3D RT calculations. By default, the
maximum number of CPUs will be used. Since EaR³T is designed to make the process of setting
up and running 3D-RT calculations simple, some parameters that are unavailable from the input
data but are required by the RT solvers are populated via default values and assumptions. However,
this does not mean that by using EaR³T, one must use these assumptions; they can be easily
superseded by user-provided settings. To facilitate this process, Table A1 provides a detailed list
of parameters (subject to change in future updates) that can be controlled and modified by the user.
In `examples/02_modis_rad-sim.py`, we defined these user-controllable parameters as
global variables for providing easy access to user. In the future, most of the parameters will be
controllable through a dedicated configuration file for optimal transparency. These parameters can
be changed within the code. For instance, by changing the parameters of `'date'` (Line 67 in
`examples/02_modis_rad-sim.py`)      and      `'region'`      (Line      68      in
`examples/02_modis_rad-sim.py`) within `params` into the following:
`params['date']   = datetime.datetime(2022, 2, 10)`
`params['region'] = [-6.8, -2.8, 17.0, 21.0]`
one can perform similar RT calculations (as demonstrated in App. 2) for another date and region
of interest (here, west Sahara Desert on 10 February, 2022). Note that the code is under active
development, the line numbers are only valid in the version release of v0.1.1 and might change in
the future. Given the input parameters, EaR³T will calculate radiance or irradiance and save the
calculations into a HDF5 (Hierarchical Data Format version 5) file. The output data variables are
provided in Table A2.
In addition to the example code, intuitive and simple examples are provided in
`examples/00_er3t_mca.py` and `examples/00_er3t_lrt.py` for users who are
interested in learning the basics of setting up EaR³T for calculations. At the current stage, only
limited documentation is provided. However, community support is available from the author of
this paper through Discord[6]. In the near-future, more effort will be invested into documentation to
give the user more autonomy in creating new applications that cannot be derived from those
provided in our paper.

| Parameters | App. 1 `examples/01_oco2_rad-sim.py` | App. 2 `examples/02_modis_rad-sim.py` | App. 3 `examples/03_spns_flux-sim.py` | App. 4 `examples/04_cam_nadir_rad-sim.py` | App. 5 `examples/05_cnn-les_rad-sim.py` |
|---|---|---|---|---|---|
| Date | September 2, 2019<br><br>Specified at Line 66: `params['date']` And Line 1569: `date` | September 2, 2019<br><br>Specified at Line 68: `params['date']` And Line 1311: `date` | September 20, 2019<br><br>Specified at Line 439: `date` And Line 238: `date` | October 5, 2019<br><br>Specified at Line 59: `params['date']` And Line 215: `date` | October 5, 2019<br><br>Specified at Line 58: `params['date']` And Line 126: `date` |
| Geographical Region | Specified at Line 69: `params['region']` | Specified at Line 69: `params['region']` | Variable (depends on aircraft location) | N/A | N/A |
| Z Grid (Number of Grids/Resolution) | 40 / 0.5 km<br><br>Specified at Line 1476: `levels` | 40 / 0.5 km<br><br>Specified at Line 1220: `levels` | 20 / 1 km<br><br>Specified at Line 180: `levels` | 40 / 0.5 km<br><br>Specified at Line 174: `levels` | 50 / 0.4km<br><br>Specified at Line 92: `levels` |
| Wavelength | 768.52 nm<br><br>Specified at Line 67: `params['wavelength']` | 650 nm<br><br>Specified at Line 67: `params['wavelength']` | 745 nm<br><br>Specified at Line 440: `wavelength` | 600 nm<br><br>Specified at Line 58: `params['wavelength']` | 600 nm<br><br>Specified at Line 57: `params['wavelength']` |
| Atmospheric Gas Profile | US standard atmosphere<br><br>Specified at Line 1479: `atm0` | US standard atmosphere<br><br>Specified at Line 1223: `atm0` | US standard atmosphere<br><br>Specified at Line 183: `atm0` | US standard atmosphere<br><br>Specified at Line 177: `atm0` | US standard atmosphere<br><br>Specified at Line 68: `params['atmospheric_profile']` And Line 94: `atm0` |
| Atmospheric Gas Absorption | Case specific<br><br>Specified at Line 1487: `abs0` | Default Absorption Database (Coddington et al., 2008)<br><br>Specified at Line 1230: `abs0` | Default Absorption Database (Coddington et al., 2008)<br><br>Specified at Line 189: `abs0` | Default Absorption Database (Coddington et al., 2008)<br><br>Specified at Line 184: `abs0` | Default Absorption Database (Coddington et al., 2008)<br><br>Specified at Line 97: `abs0` |
| Cloud Top Height (CTH) | From MODIS L2 cloud product<br><br>Specified at Line 1520: `data['cth_2d']` And Line 1530: `cld0` | From MODIS L2 cloud product<br><br>Specified at Line 1263: `data['cth_2d']` And Line 1273: `cld0` | From AHI L2 cloud product<br><br>Specified at Line 208: `cth_2d` And Lines 212: `cld0` | 2 km<br><br>Specified at Line 63: `params['cloud_top_height']` And Lines 199: `cld0` | From LES<br><br>Specified at Line 103: `cld0` |
| Cloud Geometrical Thickness | 1 km for CTH < 4 km; Variable that cloud base height is at 3 km for CTH > 4 km<br><br>Specified at Line 1527: `cgt` | 1 km for CTH < 4 km; Variable that cloud base height is at 3 km for CTH > 4 km<br><br>And Line 1270: `cgt` | 1 km<br><br>Specified at Line 212: `cgt` | 1 km<br><br>Specified at Line 64: `params['cloud_geometrical_thickness']` | From LES<br><br>Specified at Line 103: `cld0` |

---

[6] https://discord.gg/ntqsguwaWv

| | | | | | |
|---|---|---|---|---|---|
| Cloud Optical Thickness | Used IPA reflectance-to-COT mapping for MODIS L1B Reflectance at 250 m resolution<br><br>Specified at Line 1518: **data['cot_2d']** And Line 1530: **cld0** | Used IPA reflectance-to-COT mapping for MODIS L1B Reflectance at 250 m resolution<br><br>Specified at Line 1261: **data['cot_2d']** And Line 1273: **cld0** | From AHI L2 cloud product<br><br>Specified at Line 198: **cot_2d** And Lines 212: **cld0** | Used IPA reflectance-to-COT mapping and CNN for camera red channel radiance/reflectance at 100 m resolution<br><br>Specified at Lines 474 and 493: **cot_2d** And Lines 199: **cld0** | From LES<br><br>Specified at Line 103: **cld0** |
| Cloud Effective Radius | From MODIS L2 Cloud Product<br><br>Specified at Line 1519: **data['cer_2d']** And Line 1530: **cld0** | From MODIS L2 Cloud Product<br><br>Specified at Line 1262: **data['cer_2d']** And Line 1273: **cld0** | From AHI L2 cloud product<br><br>Specified at Line 199: **cer_2d** And Lines 212: **cld0** | 12 micron<br><br>Specified at Lines 475 and 494: **cer_2d** And Lines 199: **cld0** | From LES<br><br>Specified at Line 103: **cld0** |
| Scattering Phase Function | Mie (water cloud)<br><br>Specified at Line 1536 **pha0** And Line 1573: **sca** | Mie (water cloud)<br><br>Specified at Line 1279: **pha0** And Line 1315: **sca** | Mie (water cloud)<br><br>Specified at Line 219: **pha0** And Line 237: **sca** | Mie (water cloud)<br><br>Specified at Line 190: **pha0** And Line 219: **sca** | Mie (water cloud)<br><br>Specified at Line 111: **pha0** And Line 130: **sca** |
| Surface Albedo | From MODIS surface albedo product and scaled by OCO-2<br><br>Specified at Line 1501: **mod43** And Line 1503: **sfc_2d** | From MODIS surface albedo product<br><br>Specified at Line 1244: **mod43** And Line 1246: **sfc_2d** | 0.03<br><br>Implicitly specified by default at Line 234: **mcarats_ng** | 0.03<br><br>Specified at Line 61: **params['surface_albedo']** And Line 218: **surface_albedo** | 0.03<br><br>Specified at Line 59: **params['surface_albedo']** And Line 133: **surface_albedo** |
| Solar Zenith Angle | From OCO-2 geolocation file<br><br>Specified at Line 1554: **sza** And Line 1576: **solar_zenith_angle** | From MODIS geolocation file<br><br>Specified at Line 1296: **sza** And Line 1318: **solar_zenith_angle** | Variable (depends on aircraft location and date and time) | 28.90°<br><br>Specified at Line 464: **geometry['sza']** And Line 222: **solar_zenith_angle** | 29.16°<br><br>Specified at Line 60: **params['solar_zenith_angle']** And Line 134: **solar_zenith_angle** |
| Solar Azimuth Angle | From OCO-2 geolocation file<br><br>Specified at Line 1555: **saa** And Line 1577: **solar_azimuth_angle** | From MODIS geolocation file<br><br>Specified at Line 1297: **saa** And Line 1319: **solar_azimuth_angle** | Variable (depends on aircraft location and date and time) | 296.83°<br><br>Specified at Line 465: **geometry['saa']** And Line 223: **solar_azimuth_angle** | 296.83°<br><br>Specified at Line 61: **params['solar_azimuth_angle']** And Line 135: **solar_azimuth_angle** |
| Sensor Altitude | 705 km (satellite altitude)<br><br>Implicitly specified by default at Line 1568: **mcarats_ng** | 705 km (satellite altitude)<br><br>Implicitly specified by default at Line 1310: **mcarats_ng** | N/A, three-dimensional irradiance outputs at user-defined Z grid | 5.48 km (flight altitude)<br><br>Specified at Line 466: **geometry['alt']** And Line 224: **sensor_altitude** | 705 km (satellite altitude)<br><br>Specified at Line 64: **params['sensor_altitude]** And Line 138: **sensor_altitude** |
| Sensor Zenith Angle | From OCO-2 geolocation file<br><br>Specified at Line 1557: **vza** | From MODIS geolocation file<br><br>Specified at Line 1302: **vza** | 0° (nadir)<br><br>Implicitly specified by default at Line 234: **mcarats_ng** | 0° (nadir)<br><br>Implicitly specified by default at Line 214: **mcarats_ng** | 0° (nadir)<br><br>Specified at Line 62: **params['sensor_zenith_angle']** |

| | | | | | |
|---|---|---|---|---|---|
| | And Line 1578:<br>**sensor_zenith_angle** | And Line 1320:<br>**sensor_zenith_angle** | | | And Line 136:<br>**sensor_zenith_angle** |
| Sensor Azimuth Angle | From OCO-2 geolocation file<br><br>Specified at Line 1558: **vaa**<br>And Line 1579:<br>**sensor_azimuth_angle** | From MODIS geolocation file<br><br>Specified at Line 1303: **vaa**<br>And Line 1321:<br>**sensor_azimuth_angle** | 0° (insignificant for nadir)<br><br>Implicitly specified by default at Line 234:<br>**mcarats_ng** | 0° (insignificant for nadir)<br><br>Implicitly specified by default at Line 214:<br>**mcarats_ng** | 0° (insignificant for nadir)<br><br>Specified at Line 63:<br>**params['sensor_azimuth_angle']**<br>And Line 137:<br>**sensor_azimuth_angle** |
| Number of Photons | $1\times10^8$ per run<br><br>Specified at Line 70:<br>**params['photon']**<br>And Line 1583:<br>**photons** | $1\times10^8$ per run<br><br>Specified at Line 70:<br>**params['photon']**<br>And Line 1325:<br>**photons** | $1\times10^7$ per run<br><br>Specified at Line 50:<br>**params['photon']**<br>And Line 243:<br>**photons** | $1\times10^7$ per run<br><br>Specified at Line 60:<br>**params['photon']**<br>And Line 228:<br>**photons** | $1\times10^8$ per run<br><br>Specified at Line 65:<br>**params['photon']**<br>And Line 141:<br>**photons** |
| Number of Runs | 3<br><br>Specified at Line 1581: **Nrun** | 3<br><br>Specified at Line 1323: **Nrun** | 3<br><br>Specified at Line 242: **Nrun** | 3<br><br>Specified at Line 226: **Nrun** | 3<br><br>Specified at Line 140: **Nrun** |
| Mode (3D or IPA) | 3D and IPA<br><br>Specified at Line 1704 and 1705: **solver**<br>And Line 1584: **solver** | 3D or IPA<br><br>Specified at Line 1418: **solver**<br>And Line 1326: **solver** | 3D and IPA<br><br>Specified at Lines 377 and 378: **solver**<br>And Line 244: **solver** | 3D<br><br>Specified at Lines 507 and 508: **solver**<br>And Line 229: **solver** | 3D<br><br>Specified at Line 143: **solver** |
| Parallelization Mode | Python multi-processing<br><br>Specified at Line 1586: **mp_mode** | Python multi-processing<br><br>Specified at Line 1328: **mp_mode** | Python multi-processing<br><br>Specified at Line 247: **mp_mode** | Python multi-processing<br><br>Specified at Line 231: **mp_mode** | Python multi-processing<br><br>Specified at Line 145: **mp_mode** |
| Number of CPUs | 12<br><br>Specified at Line 71: **params['Ncpu']**<br>And Line 1585: **Ncpu** | 12<br><br>Specified at Line 71: **params['Ncpu']**<br>And Line 1327: **Ncpu** | 12<br><br>Specified at Line 311: **Ncpu**<br>And Line 246: **Ncpu** | 12<br><br>Specified at Line 230: **Ncpu** | 24 on clusters<br><br>Specified at Line 144: **Ncpu** |


**Table A1**: List of parameters used in the five applications. The line numbers used in the table are referring to the code
script of each application. If two line numbers are provided, the first one indicates where the parameter is
defined and the second one indicates where the parameter is passed into the radiative transfer setup. Users
can change either one for customization purposes.


| Metadata | | | |
|---|---|---|---|
| Variable Name | Description | Data Type | Dimension |
| mean/N_photon | Number of photons per run | Array | N_g |
| mean/N_run | Number of runs | Integer value | N/A |
| mean/toa | TOA downwelling flux | Float value | N/A |
| **Radiance** | | | |
| Variable Name | Description | Data Type | Dimension |

| Variable Name | Description | Data Type | Dimension |
|---|---|---|---|
| `mean/rad` | Radiance field at user specified altitude averaged over different runs | Array | (N_x, N_y) |
| `mean/rad_std` | Standard deviation of the radiance fields from different runs | Array | (N_x, N_y) |
| **Irradiance** | | | |
| Variable Name | Description | Data Type | Dimension |
| `mean/f_down` | Downwelling irradiance averaged over different runs | Array | (N_x, N_y, N_z) |
| `mean/f_down_std` | Standard deviation of the downwelling irradiance from different runs | Array | (N_x, N_y, N_z) |
| `mean/f_down_diffuse` | Diffuse downwelling irradiance averaged over different runs | Array | (N_x, N_y, N_z) |
| `mean/f_down_diffuse_std` | Standard deviation of the diffuse downwelling irradiance from different runs | Array | (N_x, N_y, N_z) |
| `mean/f_down_direct` | Direct downwelling irradiance averaged over different runs | Array | (N_x, N_y, N_z) |
| `mean/f_down_direct_std` | Standard deviation of the direct downwelling irradiance from different runs | Array | (N_x, N_y, N_z) |
| `mean/f_up` | Upwelling irradiance averaged over different runs | Array | (N_x, N_y, N_z) |
| `mean/f_up_std` | Standard deviation of the upwelling irradiance from different runs | Array | (N_x, N_y, N_z) |


**Table A2**: Data variables contained in the output HDF5 file from EaR³T for radiance and irradiance calculations. The radiance is simulated with a user-specified sensor geometry at a given altitude using forward photon tracing. The data variables listed under Metadata are included for both radiance and irradiance calculations. N_x, N_y, and N_z are the number of pixels along x, y, and z direction, respectively. N_g is the number of g, explained in Appendix A2 – Correlated-k.

**A2 – EaR³T Code Walk-through**

We will provide a code walk-through of the OCO-2 and MODIS simulator applications with the codes `examples/01_oco2_rad-sim.py` (App. 1) and `examples/02_modis_rad-sim.py` (App. 2). The data acquisition (first step in Figure 1)

uses functions in `er3t/util`. App. 1 and App. 2 use the functions in `er3t/util/modis.py`
and `er3t/util/oco2.py` for downloading the MODIS and OCO-2 data files from the
respective NASA data archives and for processing the data (e.g., geo-mapping, gridding etc.). The
user supplies minimum input (date and time, as well as latitudes and longitudes of the region of
interest), which need to be specified in `satellite_download` (within the application codes).
For example, for App. 1 and App. 2, the only user inputs are the date and time and the region of
interest – in this case September 2, 2019, with the westernmost, easternmost, southernmost, and
northernmost longitudes and latitudes of 109°W, 107°W, 37°N, and 39°N. In order for EaR³T to
access any data archives such as NASA Earthdata, the user needs to create an account with them
and store the credentials locally (detailed instructions are provided separately along with the EaR³T
distribution).

After the data acquisition step, the satellite data are fed into the pre-processing step for 1)
atmospheric gases (`er3t/pre/atm`), 2) clouds (`er3t/pre/cld`), 3) surface
(`er3t/pre/sfc`) as shown in Figure 1. In the default configuration of the App. 1, the standard
US atmosphere (Anderson et al., 1986; included in the EaR³T repository) is used within `atm`.
EaR³T supports the input of user-specified atmospheric profiles, e.g., atmospheric profiles from
reanalysis data for App. 2, by making changes in `atm_atmmod` (from `er3t/pre/atm`).
Subsequently, molecular scattering coefficients are calculated by `cal_mol_ext` (from
`er3t/util`), and absorption coefficients for atmospheric gases are generated by
(`er3t/pre/abs`). At the current development stage, two options are available:
1. Line-by-line (used by App. 1): The repository includes a sample file of absorption coefficient
profiles for a subset of wavelengths within OCO-2's Oxygen A-Band channel, corresponding
to a range of atmospheric transmittance values from low (opaque) to high (so-
called "continuum" wavelength). They were generated by an external code based on OCO-
2's line-by-line absorption coefficient database (ABSCO, Payne et al., 2020). They are
calculated for a fixed mixing ratio of 400 ppm. In a subsequent paper, an OCO-2 specific
EaR³T code will be published where the actual mixing ratio is used. For each OCO-2
spectrometer wavelength within a given channel, hundreds of individual absorption
coefficient profiles at the native resolution of ABSCO need to be considered across the
instrument line shape (ILS, also known as the slit function) of the spectrometer. The ILS, as
well as the incident solar irradiance, are also included in the file. In subsequent steps, EaR³T

performs RT calculations at the native spectral resolution of ABSCO, but then combines the

output by convolving with the ILS and outputs OCO-2 radiances or reflectances at the subset

of wavelengths. For probabilistic (Monte Carlo) RT solvers such as MCARaTS, the number

of photons can be kept relatively low (e.g., $10^6$ photons), and can be adjusted according to

the values of the ILS at a particular ABSCO wavelength. Any uncertainty at the ABSCO

spectral resolution due to photon noise is greatly reduced by convolving with the ILS for the

final output.

2. Correlated-k (used by App. 2): This approach (Mlawer et al., 1997) is appropriate for

instruments such as MODIS with much coarser spectral resolution than OCO-2, as well as

for broadband calculations. In contrast to the line-by-line approach, RT calculations are not

performed at the native resolution of the absorption database, but at Gaussian quadrature

points (called "g's") that represent the full range of sorted absorption coefficients, and then

combined using Gaussian quadrature weights. The repository includes an absorption

database from Coddington et al. (2008), developed specifically for a radiometer with

moderate spectral resolution on the basis of HITRAN (high-resolution transmission

molecular absorption database) 2004 (Rothman et al., 2005). It was created for the ILS of

the airborne Solar Spectral Flux Radiometer (SSFR, Pilewskie et al., 2003), but is applied to

MODIS here, which has a moderate spectral resolution of 8-12 nm with 20-50 nm

bandwidths. It uses 16 absorption coefficient bins (g's) per target wavelength (this could

either be an individual SSFR or a MODIS channel), which are calculated by EaR$^3$T with the

Coddington et al. (2008) database using the mixing ratios of atmospheric gases in the

previously ingested profile. In future implementations, the code will be updated to enable

flexible ILS and broadband calculations.

The `er3t/pre/cld` module calculates extinction, thermodynamic phase, and effective

droplet radius of clouds from the input data. The `er3t/pre/pha` module creates the required
single scattering albedo and scattering phase function. The default is a Henyey-Greenstein phase
function with a fixed asymmetry parameter of 0.85. Along with the current distribution (v0.1.1) of
EaR$^3$T, the Mie phase functions based on thermodynamic phase, effective droplet radius, and
wavelength are supported. In this study, App. 1 and App. 2 use Mie phase functions calculated
from Legendre polynomial coefficients (originally distributed along with libRadtran) based on the
wavelength and cloud droplet effective radius. In the future, EaR$^3$T will include stand-alone phase
functions, which can be chosen on the basis of droplet size distributions in addition to effective
radius. It is also possible to include aerosols in a similar fashion as clouds. This is done with the
`er3t/pre/aer` module. In the case of aerosols, spectral single scattering albedo and asymmetry
parameter are required as inputs in addition to the extinction fields.

After the optical properties are calculated, they are passed into the 3D-RT step

(`er3t/rtm/mca`). This step performs the setup of RT solver-specified input parameters and data
files, distributing runs over multiple Central Processing Units (CPUs), and post-processing RT
output files into a single, user-friendly HDF5 file. For example, when radiance is specified as
output (default in App. 1 and App. 2), key information such as the radiance field and its standard
deviation are stored in the final HDF5 file (details see Table 1).

While the EaR$^3$T repository comes with various applications such as App. 1 and App. 2,

described above, the functions used by these master or 'wrapper' programs can be organized in
different ways, where the existing applications serve as templates for a quick start when developing
new applications. The functions used by the master code pass information through the various
steps as Python objects. For example, in `examples/01_oco2_rad-sim.py`, the downloaded
and processed satellite data are stored into the `sat` object. Later, the `sat` object is passed into an
EaR$^3$T function to create the `cld` object that contains cloud optical properties. Similarly, EaR$^3$T
provides functions to create the `atm`, and `sfc` objects with optical properties for atmospheric
gases and the surface. These objects (`atm`, `cld`, `sfc`) are in turn passed on to solver-specific
modules for performing RT calculations. The user can choose to save the data of the intermediate
objects into Python pickle files after the first run. In this way, multiple calls with identical input
can re-use existing data, which accelerates the processing time of EaR$^3$T. Unless the user specifies
the `overwrite` keyword argument in the object call to reject saving pickle files, these shortcuts
save significant time.

**Appendix B – App. 5 Radiance calculations based on the Large Eddy Simulation**

The CNN COT retrieval framework was developed by Nataraja et al. (2022). It adapts a

U-Net (Ronneberger et al., 2015) architecture and treats the retrieval of COT from radiance as a
segmentation problem – probabilities of 36 COT classes (ranging from COT of 0 to 100) are
returned as the final COT retrieved for a given cloud radiance field. It accounts for horizontal
photon transport, which is neglected in traditional cloud retrieval algorithms; in other words, for
the spatial context of cloudy pixels. It was trained on synthetic cloud fields generated by a Large
Eddy Simulation (LES) model, which provides the ground truth of COT. Subsequently, EaR[3]T was
used to calculate 3D-RT radiances at 600 nm for LES cloud fields to establish a mapping between
radiance to COT. Only six LES cases were used to represent the variability of the cloud
morphology. Each of these fields are 480x480 pixels across (spatial resolution of 100 m). These
large fields were mapped onto thousands of 64x64 mini tiles with spatial resolution of 100 m as
described in Nataraja et al., 2022. To keep the training data set small, mini tiles selectively sampled
according to their mean COT and standard deviation. This ensured an even representation of the
dynamic range of COT and its variability, which was termed homogenization of the training data
set. Figure A1 shows a collection of samples from the training data as an illustration. All the
aforementioned simulation setup and techniques in data process are included in the App. 5 example
code, which can be applied to the LES data (a different scene from the 6 scenes) distributed along
with EaR[3]T.


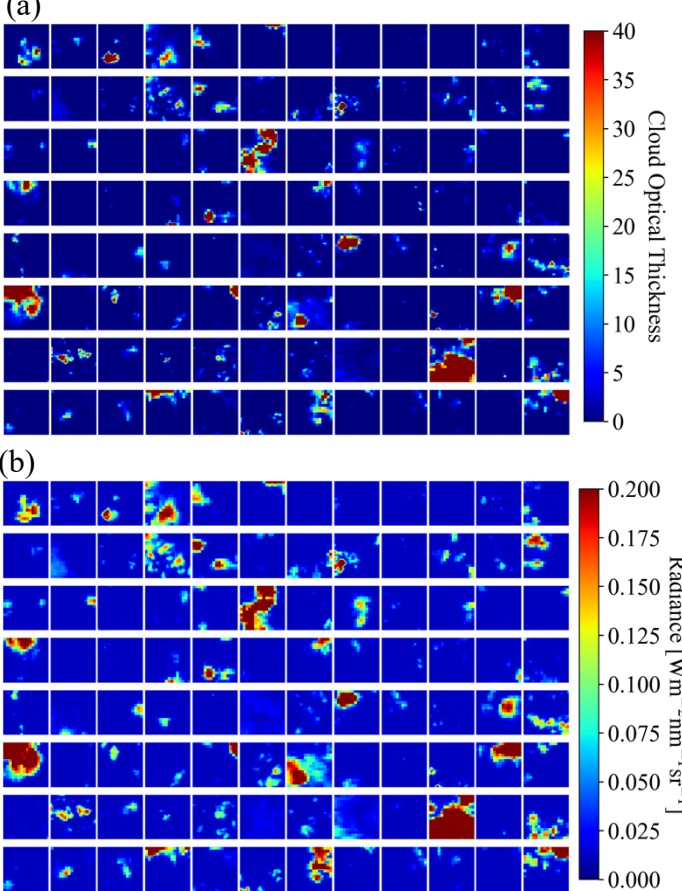


**Figure A1**. Illustrations of 64x64 tiles of **(a)** cloud optical thickness from LES data and **(b)** calculated 3D radiance at
600 nm from EaR³T for CNN training.


**Appendix C**
**C1. Cloud Detection/Identification**
Cloudy pixels are identified through a thresholding method based on the red, green, and
blue channels of MODIS. When the radiance values of the red, green, and blue channels of a pixel
are all greater than a pre-calculated threshold value, the pixel is considered as cloudy, as illustrated
by the following equation
**If** $\begin{array}{l} Red > a_R \cdot Quantile(Red, q_0) \ \& \\ Blue > a_B \cdot Quantile(Blue, q_0) \ \& \\ Green > a_G \cdot Quantile(Green, q_0) \end{array}$ $\begin{cases} \textbf{Yes}: \text{cloudy} \\ \textbf{No}: \text{clear sky} \end{cases}$     (A1)
where $a_R$, $a_B$, and $a_G$ are scale factors with a default value of 1.0, and $Quantile$ returns the $q_0$
percentile of the sorted reflectance data (ascending order; $q_0 = 0.5$ is equivalent to the median).
The scale factors can be adjusted separately to perform fine tuning for different surface types. For
example, adjusting $a_G$ will be more effective for separating clouds from greenish vegetation
surface than the other two factors. For simplicity, they are all set to 1.0 for the case shown in App.
1 and 2. The $q_0$ is determined by the following equation,
$q_0 = \max(0, \quad 1 - frac_{cld} \cdot 1.2)$     (A2)
where $frac_{cld}$ is cloud fraction obtained from the MODIS L2 cloud product (number of cloudy
pixels divided by the number of total pixels). Through the definition of $q_0$, the threshold-based
cloud detection method is pegged to the MODIS product at the domain scale. Because of the coarse
resolution of the MODIS-based cloud mask, it cannot be used directly for our application.
However, it uses many more channels than available at high spatial resolution, and is therefore
more accurate. The factor of 1.2 can be adjusted. A value of higher than 1 allows for clouds that
are not detected by MODIS (for various reasons, for example because of their spatial scale) to be
picked up. At the same time, this leads to over-detection (false positives, i.e. clear-sky pixels
identified as cloudy), and therefore the thresholding is only the first step (primary thresholding),
followed by the next (secondary) step where false positives are removed.
The secondary step is based on MODIS L2 cloud products: *COT* (cloud optical thickness),
*CER* (cloud effective radius), and *CTH* (cloud top height). For the pixels that are identified as
cloudy in the primary thresholding, especially at the lower end of the reflectance (*Ref.*), we rely
on the clear-sky identifiers from MODIS L2 cloud product (where no cloud products are retrieved),
as illustrated by the following equation
$\quad\textbf{If}\quad$ $\begin{array}{l}Ref. < Median(Ref.)\ \&\\ COT, CER, and\ CTH\ are\ NaN\end{array}$ $\left\{\begin{array}{l}\textbf{Yes}: \text{clear sky}\\ \textbf{No}: \text{cloudy}\end{array}\right.$ $\qquad$ (A3)
Figure A2 shows the cloud mask from primary thresholding (Equation A1, red and purple), and
the pixels that are reverted to clear-sky by the secondary filter (Equation A2, red).

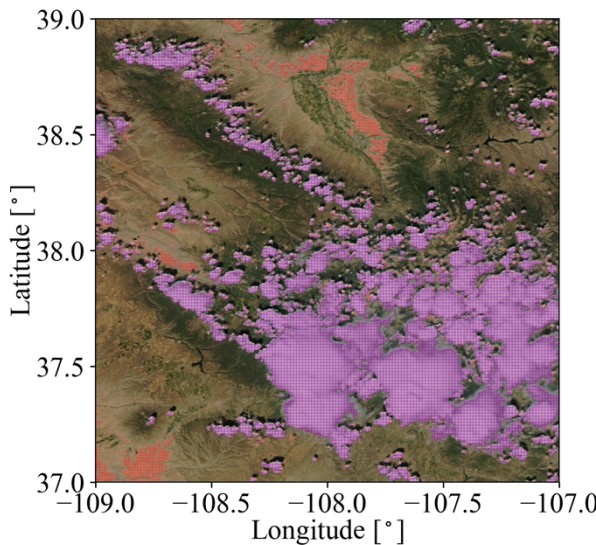


**Figure A2**. Cloud mask for the scene shown in Figure 2. Red and purple indicate pixels identified as cloudy through
$\qquad$ the primary thresholding (Equation A1) and purple indicates pixels finally identified as cloudy after applying
$\qquad$ secondary filter (Equation A3).

**C2. IPA Reflectance-to-COT Mapping**
$\qquad$ In order to retrieve COT (cloud optical thickness) from cloud reflectance as measured by
various instruments, we use the EaR$^3$T built-in solver MCARaTS in IPA mode to calculate a
lookup table of reflectance as a function of COT. The function for generating these lookup tables
is included in EaR$^3$T as `er3t.rtm.mca.func_ref_vs_cot`. Two mappings are generated
for App. 1&2 to account for geometrically thin (cloud top height less than 4 km) and thick (cloud
top height greater than 4 km) clouds separately while a single mapping is generated for App. 4.
Specifically, for a range of COT (0 to 200), reflectance is calculated from EaR$^3$T with the same
input parameters (wavelength, viewing and solar geometries, and surface albedo) listed in Table
A1 for each application except for a few simplifications described in the following table (Table
A3):

| | App. 1 & 2 | | App. 4 |
|---|---|---|---|
| Cloud Type | Geometrically Thin Clouds | Geometrically Thick Clouds | All |
| Cloud Effective Radius | 10 $\mu m$ | 20 $\mu m$ | 10 $\mu m$ |
| Cloud Top Height | 3 km | 10 km | 2 km |
| Cloud Geometrical Thickness | 1 km | 7 km | 1 km |
| Surface Albedo | 0.08 (domain average of the MCD43 WSA) | 0.08 (domain average of the MCD43 WSA) | 0.03 |


**Table A3**: List of parameters for deriving IPA reflectance-to-COT (cloud optical thickness) mappings for App. 1&2
and App. 4 in addition to Table A1.

The clouds are assumed horizontally homogeneous over a $2 \times 2$ pixel domain. For each
calculation, $10^8$ photons are used for running EaR$^3$T in IPA mode. After calculating $R(COT)$, the
inverse relationship of $COT(R)$ is then used for estimating $COT$ at any given $R$ for the cloudy
pixels. Figure A3 shows the IPA reflectance-to-COT mappings created for App. 1&2, and App 4.
Note that the difference between the App. 1&2 thin clouds (blue) and App. 4 (green) is due to
different surface albedos (when COT less than 20) and sensor viewing geometries (when COT
greater than 20, specified in Table A1). Note that this approach will ensure IPA
radiance/reflectance consistency (retrieved IPA COT will reproduce the exact IPA cloud
reflectance, see Figure A4) because the radiative transfer processes of $R(COT)$ and $COT(R)$ are
the same. However, since it makes some simplifications as mentioned above, uncertainties are
expected for a complicated atmospheric environment (varying cloud thermodynamic phase,
effective radius, cloud top height, geometrical thickness, vertical profile; variable surface albedo
and topography), which are shown up as spread (deviations from identity line) in Figure A4.

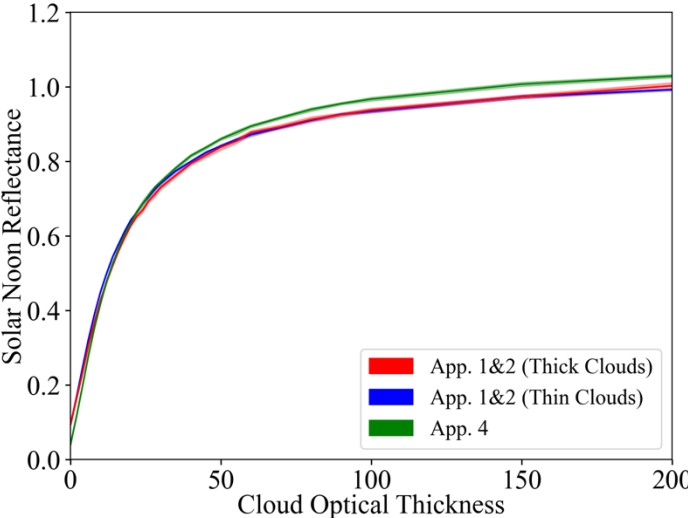


**Figure A3**. The IPA reflectance-to-COT mappings used for App. 1&2 (red and blue) and App. 4 (green). The reflectance is normalized by the cosine of solar zenith angle (referred to as solar noon reflectance). The uncertainties associated with photon statistics are indicated by the shaded area.


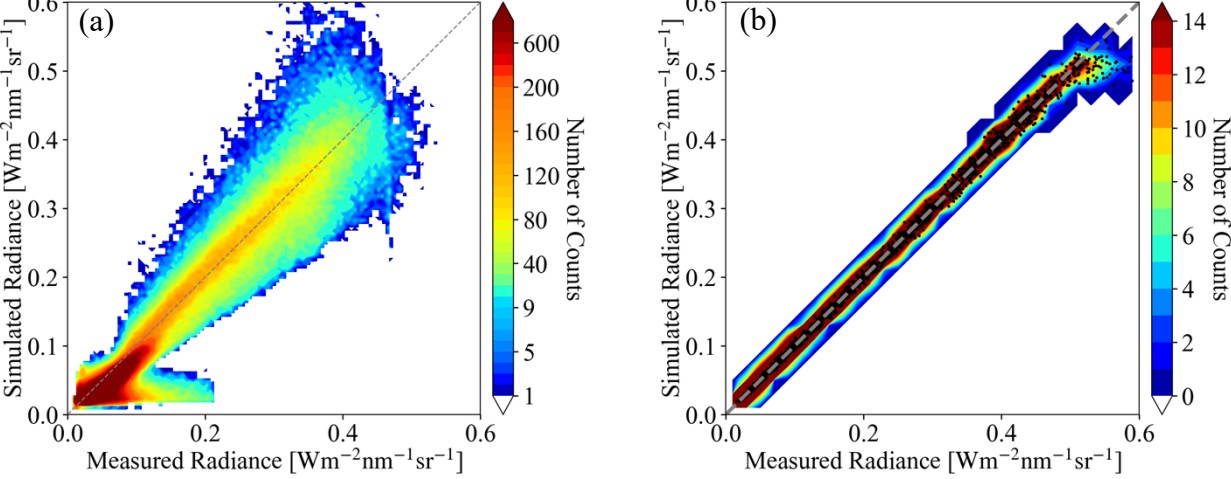


**Figure A4. (a)** and **(b)** are the same as Figure 7 and Figure 13b except for the IPA radiance calculations.



## Appendix D

### D1. Parallax Correction

From the satellite's view, the clouds (especially high clouds) will be placed at inaccurate locations on the surface, which have shifted from their actual locations due to the parallax effect. We followed simple trigonometry to correct for it, as follows:

Longitude correction (positive from west to east):

$$\delta lon = \frac{(z_{cld} - z_{sfc}) \cdot \tan(\theta) \cdot \sin(\phi)}{\pi \cdot R_{Earth}} \times 180° \tag{A4}$$
Latitude correction (positive from south to north):
$$\delta lat = \frac{(z_{cld} - z_{sfc}) \cdot \tan(\theta) \cdot \cos(\phi)}{\pi \cdot R_{Earth}} \times 180° \tag{A5}$$
where $(lon_{sat}, lat_{sat}, z_{sat})$ is the satellite location and $\theta$ and $\phi$ (0° at north, positive clockwise)
are the sensor viewing zenith and azimuth angles. $z_{cld}$ and $z_{sfc}$ are the cloud top height and the
surface height. $R_{Earth}$ is the radius of the Earth. Figure A2 shows an illustration of the parallax
correction for the cloud field in the inset in Figure 2. Note that discontinuities in the latitude and
longitude fields arising from different combinations of sensor viewing geometries and cloud top
and surface heights may lead to gaps in the cloud fields. These gaps are identified and filled in
with the average of data from adjacent pixels (plus minus two pixels along x and y) through the
following process:
**If** $\begin{array}{l} pixel_{ij}^{aft} \text{ is clear } \& pixel_{ij}^{bef} \text{ is cloudy } \& \\ cldfrac(pixel^{bef}[i-2:i+2, j-2:j+2]) > frac_a \& \\ cldfrac(pixel^{aft}[i-2:i+2, j-2:j+2]) > frac_b \& \end{array}$ $\begin{cases} \textbf{Yes}: \text{fill } pixel_{ij}^{aft} \text{ with the average of} \\ cld(pixel^{aft}[i-2:i+2, j-2:j+2]) \end{cases}$
where $pixel_{ij}$ indicates the pixel at $i$ along x and $j$ along y, $bef$ and $aft$ refer to before and after
parallax correction respectively, $cldfrac$ calculates cloud fraction (number of cloudy pixels
divided by total pixel number), and $cld$ selects data where pixels are identified as cloudy. The
$frac_a$ and $frac_b$ are set to 0.7 for the cases demonstrated in the paper. Lower $frac_a$ tends to over
select clear-sky pixels at the cloud edge and lower $frac_b$ tends to over correct clear-sky pixels
within clouds that are not clear-sky due to parallax artifacts. While increase $frac_a$ and $frac_b$
tends to under correct parallax artifacts.

**D2. Wind Correction**
The wind correction aims at correcting the movement of clouds when advected by the wind
between two different satellites' overpasses.
Longitude correction (positive from west to east):
$$\delta lon = \frac{\bar{u} \cdot \delta t}{\pi \cdot R_{Earth}} \times 180° \tag{A6}$$
Latitude correction (positive from south to north):

$$\delta lat = \frac{\bar{v} \cdot \delta t}{\pi \cdot R_{Earth}} \times 180° \qquad (A7)$$

where $\bar{u}$ and $\bar{v}$ are the domain-averaged 10 m zonal and meridional wind speeds, and $\delta t$ is the time difference between two different satellites that fly on the same orbit. Figure A2 shows the cloud location after applying the parallax (Appendix D1) and wind correction for the cloud field in the inset from Figure 2.

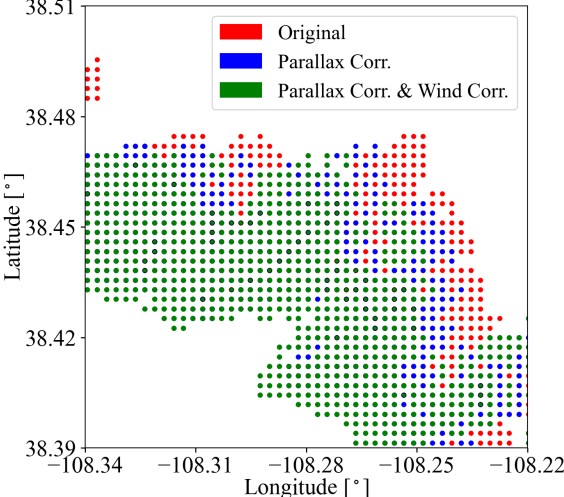

**Figure A5**. An illustration of correcting cloud location (red) for parallax effect (blue) and wind effect (green) for the cloud field of the inset in Figure 2. Filled cloud gaps as described in Appendix D1 are indicated by black circles.

**Acknowledgement**

The aircraft all-sky camera was radiometrically calibrated by the U.S. Naval Research Laboratory. We thank Jens Redemann for insightful discussions on Figure 9 (App. 3) about the apparent contradiction of the direction of the COT, reflectance, and transmittance biases.

**Data availability**

For App. 1 and App. 2, the OCO-2 data were provided by the NASA Goddard Earth Sciences Data and Information Services Center (GES DISC, https://oco2.gesdisc.eosdis.nasa.gov/data) and the MODIS data were provided by the NASA Goddard Space Flight Center's Level-1 and Atmosphere Archive and Distribution System (LAADS, https://ladsweb.modaps.eosdis.nasa.gov/archive), which are all publicly available and can be downloaded by EaR$^3$T through the application code. For App. 3, the AHI data were processed by Holz's (coauthor of this paper) team. The SPN-S data were provided by Schmidt and Norgren (coauthors of this paper). Both the AHI and SPN-S data are publicly available at NASA Airborne Science Data for Atmospheric Composition (https://www-air.larc.nasa.gov/missions/camp2ex/index.html). The AHI data and the SPN-S data for the flight track indicated in Figure 8 of the paper are distributed along with EaR$^3$T for demonstration purpose. For App. 4, all sky camera imagery and CNN model are distributed along with EaR$^3$T. EaR$^3$T is publicly available and can be accessed and downloaded at https://github.com/hong-chen/er3t (or https://doi.org/10.5281/zenodo.7734965 for v0.1.1 used in this paper; Chen and Schmidt, 2022).

**Author contributions**

All the authors helped with editing the paper. HC developed the EaR$^3$T package in Python including the application code, performed the analysis, and wrote the majority of the paper with input from the other authors. KSS provided an initial MCARaTS simulation wrapper code in Interactive Data Language (IDL); helped with the structure design of EaR$^3$T; and helped with interpreting the results and writing the paper. SM helped with the OCO-2 data interpretation. VN trained and provided the CNN model. MN helped with the SPN-S instrument calibration and data processing. JG and GF helped with testing EaR$^3$T and the LES data interpretation. RH provided the AHI data and helped with the data interpretation. HI helped with the implementation of MCARaTS in EaR$^3$T.

**Competing Interests**

1189 K. Sebastian Schmidt is a member of the editorial board of Atmospheric Measurement Techniques.

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
