# Peer review of "The Education and Research 3D Radiative Transfer Toolbox (EaR3T) – Towards the"

_Atmospheric Measurement Techniques, 2022_

## Author Comment (AC1)

**Referee #1 (Hartwig Deneke)**

Received and published: 20 July 2022

**Note:** The page and line numbers used in the response are referring to the revised manuscript (track-change version), which is appended to this response.

**General comments**

**C:** The article describes a Python project for 3D radiative transfer, the EaR3T toolbox. While somewhat technical in scope, the article is generally well written, likely of interest to a wider scientific audience, and falls within the scope of AMT. There are however a few aspects which could be improved, which I list below. Hence, I recommend publication of the article after minor revisions.

**R:** Thank you very much for your comments.

**Specific comments**

**C:** For reproducibility, I strongly recommend to obtain a DOI for the described version of the code in the github repository, e.g. via Zenodo, see https://docs.github.com/en/repositories/archiving-a-github-repository/referencing-and-citing-content. While the article mentions "in the current version", no clear information on versioning of the code is given, this needs to be rectified, in particular, the article needs to clarify which version of the code is referred to.

**R:** Thank you very much for your suggestions and providing instructions. We have released the first official version of EaR3T (version 0.1.0) on Github (https://www.github.com/hong-chen/er3t/releases/tag/v0.1.0) and obtained DOI from Zenodo (doi:10.5281/zenodo.7374196). The information has been updated in the revised manuscript (Page 9, Line 244).

**C:** Usage of APP for application: why not App? It's used as an abbreviation, not as an acronym. **R:** We have changed the "APP" in "App." In the revised manuscript to avoid confusion between abbreviation and acronym.

**C:** As mentioned in the text, APP5 is not described, but it is included in Fig.1. I propose to also remove it from Fig.1. The description "four of which are described in this paper" at least for me raises the question why, maybe motivate this choice somewhat?

**R:** Thank you. The reason we included App. 5 in Figure 1 is that we believe context-aware CNN algorithms based on machine learning will become the key towards the mitigation of 3D cloud retrieval bias, where EaR3T shines by its automation capability of creating extensive simulation datasets for training CNN. We decided to keep App. 5 in Figure 1 and added a brief description of CNN in Appendix B with the details discussed by Nataraja et al. (2022). We hope after adding in

descriptions of CNN in the Appendix B (Page 43, Line 1174), we can keep App. 5 in Figure 1 to keep the information complete from the two papers (this paper and Nataraja et al., 2022).

**C:** Summary and Outlook: I do find the outlook somewhat too short/lacking a clear vision about future development of the code. The following sentence also raises some questions: "EaR3T will continue to be an educational tool driven by graduate students." I did not find anything indicating which parts of the code so far have been actually written by graduate students (who of the authors is at that stage?), given that several co-authors are rather senior. I also would assume that it takes someone with significant experience to maintain such a project in the long term. Please elaborate at least to some detail on these points.

**R:** Thank you. We added some text (Page 39, Line 1127) regarding future work of adding support for more publicly available 3D RT solvers, e.g., SHDOM and MYSTIC, and built-in support for HITRAN. The current version of EaR3T including code base and applications were solely developed by graduate student Hong Chen (first author of this paper) under the advisement of Prof. Sebastian Schmidt. The other authors of this paper contributed the data and model used in the four applications of EaR3T. Currently, a few other applications of EaR3T, e.g., spectral simulations for OCO-2, are under development by other graduate students in Prof. Schmidt's group. To keep the continuity, Hong Chen is committed to maintain EaR3T for the next few years and gradually transition the development and maintenance of EaR3T to other graduate students in Prof. Schmidt's group. We added "Author contribution" in the revised manuscript (Page 48, Line 1289).

C: Please also note the following minor language comments:

L264: "MODIS is currently flying on …" I doubt this will change anytime soon, rephrase sentence? **R:** We rephrased the sentence into "The MODIS instruments are multi-use multispectral radiometers onboard …" (Page 13, Line 326).

C: L265: "They are ...": Please clarify "They", I guess it refers to MODIS. R: Yes, it refers to MODIS. We changed "They are ..." to "MODIS was ..." for clarification (Page 13, Line 327).

**References:**

[revised manuscript text omitted]

---

## Author Comment (AC2)

**Referee #2 (Anonymous Referee)**

**Note:** The page and line numbers used in the response are referring to the revised manuscript (track-change version), which is appended to this response.

**Text**

**C:** The paper introduces the versatile EaR$^3$T Radiative Transfer Toolbox and showcases several applications with focus on analyzing and mitigating 3D effects in observations and simulations. I thank the authors for this overall very nice paper. I particularly find its education targeted approach very interesting. I have few, both general and specific, comments (see below) and recommend to publish the paper when these overall minor issues are addressed.
**R:** Thank you very much for your comments.

**General comments**

**C:** Some, as I understand, mandatory or highly recommended sections are missing, incl. Data availability and Author contributions.
**R:** We have added sections of data availability and author contributions into the manuscript.

**C:** As the other reviewer pointed out, it would be highly desirable if code & examples used here are available from a long-term archive.
**R:** Purposing for reproducibility and traceability, we have made a public release of EaR$^3$T (version 0.1.0) on GitHub (https://github.com/hong-chen/er3t/releases/tag/v0.1.0) and obtained DOI from Zenodo (doi: 10.5281/zenodo.7374196). The information has been updated in the revised manuscript (Page 9, Line 244).

**C:** You characterize EaR$^3$T as "automated" and running with "minimal user input" and stated that "automation of EaR$^3$T permits calculations at any time and over any region". Could you be a bit more specific, what you mean by that (automated)? On the one hand, please summarize, what user input is actually needed (what is mandatory, what optional) and how the user has to provide/setup. What does the user interface look like, how does it work? An example of a call/setup could be helpful, e.g. illustrating how the Tab2 settings (at least for one APP are realized). Also, how is the model executed; e.g. how are the steps mentioned in L425f executed? Does the user have to do that with a script, is there a ready-made script available, ...? Make clear, what is the "normal" way to do these things with EaR$^3$T.
**R:** Thank you for your comments. The regular ("normal") operation of 3D or 1D radiative transfer calculations is usually done by passing the required parameters through a parameter list contained within a formatted text or binary file along with auxiliary data in a format specific to an RT solver. This requires the user to manually adjust the input files, prepare the auxiliary data, and post process the output. The automation of EaR$^3$T minimizes these manual efforts by providing functions that can contain the parameter input within a programming language (here Python), where unlimited operations, e.g., loops, can be achieved through programming. Additionally, with the functions (offered by EaR$^3$T) that can preprocess of the auxiliary data, e.g., surface setup, clouds setup, aerosol setup etc., and postprocess of the results can be integrated, which, combined, provides automation capability. In other words, EaR$^3$T serves as a 'wrapper' to RT code, including the pre-processing and downloading of input data, and including the post-processing – but it is also more than that because it can be run along entire aircraft flight tracks or satellite orbits. The automation does not mean EaR$^3$T takes away user's freedom of specifying parameters – parameters can be specified as detailed as possible that the RT solver supports. Instead, EaR$^3$T offers flexibility to bypassing some parameters through defaults. For example, if the cloud geometric thickness is not available or specified, EaR$^3$T will use a default value (1km), which however can be controlled by the user. This allows the user to arrive at simulations without much hassle. In a way, this is similar to the RT library of libRadtran, where certain parameters can be set as "standard", and others are implicit. The input parameters that the user can control (or their defaults) were added to the manuscript for better transparency – in Appendix A in the revised manuscript. The example codes (contains interface of the code) along with guide of how to install and run are distributed along with EaR$^3$T. The example codes distributed along with EaR$^3$T are ready-to-run once setup. The example codes are designed so they can be easily adapted for related projects (e.g., for a different wavelength). Currently, the application of EaR$^3$T for completely new project that cannot be created by adapting one of the pre-existing example codes will still require some support from the author of EaR$^3$T and of this paper. This will hopefully be improved once the documentation is more complete in the future. The execution is done at the example code level, e.g., "python 02_modis_rad-sim.py" under <examples> directory will reproduce the results in Figure 6 in the manuscript, which processes involve MODIS data downloading, preprocessing of surface and cloud setup based on MODIS products, running 3D RT, and postprocessing the outputs.

**C:** On the other hand, "with minimal user input" implies that many parameter that need to be known for radiative transfer model are implicitly assumed (aka "hardcoded") - which are these and what setting/assumptions are made there? What effects do these assumptions have on the RT results?

**R:** To address this comment as well as the following few comments, we added Appendix A (Page 40, Line 1135), which provides a detailed list of controllable parameters and their default settings for each application. The effects of these assumption on the RT results are discussed in the other comments of this response.

**C:** Also, certain manual adjustments/user settings are obviously still necessary, e.g. setting of an (appropriate) SZA. I lack an overview of these additional required setup beyond time & location. Also, I'd appreciate some remarks on the ease or difficulty to adapt the examples to more or less different applications e.g. the use of different sensor channels.

**R:** The current SZA settings for App. 1 and App. 2 are using the average of SZA of observations by default. It can indeed be set differently by user. As we mentioned in our response to the previous comment, we added a detailed list of controllable parameters and their default settings for each application in Appendix A. Current example code supports simple adjustments for a slightly different application, e.g., a different wavelength of the same sensor. However, for an independent project, a different example code needs to be developed (combining the functions of EaR$^3$T in different ways) [see comment above; this will currently require support from the author].

C: One exemplary aspect here: you do not seem to consider, e.g., cloud vertical location & extent as "user input", although that needs to be specified somehow and at least partly seems to require an "educated guess" by the user, which in turn introduces a user dependent source of uncertainty.
R: In fact, EaR$^3$T can ingest the atmospheric information as detailed as the underlying RT solver can offer. For example, if cloud vertical location is available from active or passive remote sensing, EaR$^3$T can put the clouds at the correct location. If it is not known and not provided by the user, then this information is superseded by default values as explained above. However, for transparency reasons, we included a new appendix (see comment above).

C: The example scene contains different cloud types (from low to vertically extended). Here I would like to see a short discussion on the effect of fixed cloud geo thickness in the setup. How realistic is the 1km-thick-clouds setting in APP1&2 (also considering that the clouds in this scene all seem to be high-level clouds according to Fig3c (CTH>8km)? How sensitive are the outputs to this, ie what errors can result from that? How realistic is that for the cumulonimbus contained in the scene? Similarly, for APP3&4, what uncertainties do the cloud location choices induce; how sensitive are results to those choices (considering they might be off by a few km for individual clouds in a scene and that those seemingly need to be chosen by the user beforehand. what about (frequently occurring) multi-level clouds?
R: The 1km-thick-clouds assumption would only be valid for the low-level thin clouds and would not be realistic for cumulonimbus (CTH>8km). To evaluate the sensitivity to this assumption, we performed another 3D RT run with cloud base at 0.5km for all the clouds (clouds vertically extend from 0.5km to cloud top height). The radiance difference between the new run and the run with 1km-thick-clouds assumption is shown in the following Figure (Figure 6-extra-1). The figure indicates that when the clouds have a larger vertical extent, more horizontal photon transport, leading to an even larger 3D effect that for shallow clouds, i.e., smaller radiances in the core of the nimbus. Since we were looking at nadir radiance/reflectance, the radiance difference for tall vs. shallow clouds is smaller than from oblique viewing angles where the vertical structure of the clouds matters even more. We added the following discussion short discussion in the revised manuscript (Page 26, Line 734)
"It should be pointed out that the vertical extent of the clouds affects the simulated radiance – the larger the vertical extent, the larger the 3D effects (more horizontal photon transport). Since we make the assumption of a cloud geometric thickness of 1 km if no thickness information is provided, the simulated radiance at the satellite sensor level is valid for that proxy cloud only. For deeper clouds, the simulated radiance would be even lower. Either way, the comparison with the actual radiance measurements will reveal a lack of closure."
For App. 3, the uncertainties associated with 1km-thick-clouds is small because we are looking at the downwelling irradiance below clouds. For App. 4, the 1km-thick-clouds should work as the cloud location of 1km to 2km is estimated from the aircraft observations. For the applications we provided in the manuscript, the assumptions are relatively safe (nadir radiance/reflectance, transmittance, selected camera imagery). This does not mean that, when using EaR$^3$T, one has to live with assumptions. Our goal is not to fill in missing data, but use data wherever it is available, and subsequently use the radiance consistency approach to determine how accurate and appropriate these data and any assumptions were. EaR³T offers the capability to digest the details of the atmosphere if they are available, but it can also run if they are not – through the aforementioned assumptions. For global application (e.g., multi-layer clouds, aerosols, surfaces), a dedicated effort needs to be invested into developing a much more comprehensive example code than the one we discussed in the manuscript, but this is all doable using the functions provided by EaR³T. We are envisioning that with more data resources (e.g., observations from active remote sensing, 3D atmospheric retrievals (Barker et al., 2011)) and EaR³T and methodologies provided in the manuscript, we can arrive at 3D radiation closure at a global scale.

[Figure]

Figure 6-extra-1: Radiance difference between simulations using clouds 1) extending from 0.5 km to cloud top height (Radiance1) and 2) assuming clouds are 1km thick (Radiance2). The difference is Radiance1 minus Radiance2.

**C:** In Sec.3, I have some difficulties to understand, how exactly cloud extinction, phase, and Reff calculated from the input data? From which input data, specifically?
**R:** The cloud extinction, phase function, and cloud effective radius are processed during the pre-processing step. EaR³T contains functions to process these cloud properties on application basis. For example, for satellite applications, EaR³T offers functions to download the satellite data and extract cloud properties from satellite products, or directly use cloud retrievals obtained from tailored algorithms developed by user (e.g., two-stream approximation for obtaining high resolution cloud optical thickness field from radiance observations). For LES applications, EaR³T offers functions that can extract the cloud optical properties from LES data itself.

**C:** What input parameters are required for MCARaTS, specifically? How are they derived from the pre-processed cloud & surface properties – is this just a re-gridding, or does it require a(nother) parameter transformation? How would that (need to be) different for libRadtran? Some equations would be helpful here.
**R:** The MCARaTS requires the inputs of radiative properties of surface and atmospheric constituents (e.g., gases, aerosols, clouds) such as single scattering albedo, scattering phase function, or asymmetry parameters, along with solar and sensor viewing geometries. We added descriptions of input parameters for MCARaTS in the revised manuscript (Page 9, Line 245). Those optical properties are not directly contained within satellite products. The preprocessing is a parameter transformation that converts indirect atmospheric properties (e.g., cloud optical thickness and cloud effective radius) into direct optical properties of the atmospheric constituents (e.g., absorption coefficients, extinction coefficients, single scattering albedo, scattering phase function, asymmetry parameters etc.) that MCARaTS can digest. Meanwhile, the preprocessing also makes formatted input text file and auxiliary data files that contain RT parameters required by MCARaTS. For libRadtran, a different wrapper is developed within EaR$^3$T to cope with input parameters required by libRadtran in a different format than MCARaTS.

**C:** What determines the RT grid – in the horizontal as well as in the vertical? User input? Input data resolution?

R: On the input side, the vertical grid of the RT is set by the user during the atmospheric profile setup step in the pre-processing. For the vertical grid, corresponding atmospheric profile of gas concentration will be extracted (if not specified, the AFGL US standard atmosphere will be used – see Appendix A), then gas absorption coefficient profile will be extracted. Later, the clouds, aerosols will be inserted into the vertical grid through linear interpolation while the extinction coefficients will go through sum/average to ensure the column-integrated property is consistent before and after re-gridding. From the output side, the irradiance/flux will output at vertical grid set by user while the radiance will output with horizontal grid defined by the horizontal grid of cloud field.

**C:** Sort out, explain, and correctly use your terminology: Specifically, for App4 (and summary item c), disentangle IPA/2-stream vs. 3D/CNN (why does Fig11 talk about COT estimated by IPA, Fig12 of COT estimated by 2-stream? That's the same data from the same method, isn't it?).

**R:** The words of <3D and CNN> and  were used interchangeably in APP4 (COT mentioned in Figure 11 (IPA) and Figure 12 (two-stream) are the same). We changed the wording to achieve better clarification and consistency so now in APP4, we only use <CNN> and .

**C:** Avoid the impression to claim that EaR$^3$T is the only 3D-RT capable model/tool(box) (aeg around L848) – there is and has been for a long while a variety of those. libRadTran/MYSTIC, SASKTRAN (Bourassa08), McSCIA (Spada06), and SPARTA (Barlakas16) in the solar radiation part of the spectrum as well as e.g. ARTS (Buehler18) in the MW/IR region are just a few that come to mind. Also, there's the WCRP's I3RC project (Cahalan04; https://www.wcrp-climate.org/modelling-wgcm-mip-catalogue/modelling-wgcm-mips-2/261-modelling-wgcm-catalogue-i3rc). Please put your model/toolbox in context with those.

**R:** Thank you for providing the references. We briefly discussed related 3D radiative transfer solvers and toolboxes in the introduction of the revised manuscript (Page 4, Line 104), but not all of the ones that were proposed. For example, the Buehler code is applicable for the IR (to our knowledge), and our manuscript focuses on the SW.

**Specific comments**

Add axis labels incl. unit specification to all figures, and colorbars to all 2D plots.
R: We added units and/or colorbars for all the figures in the revised manuscript.

Fig1/L201f: Improve color coding explanation: I had to read a couple of time to get, e.g., the difference between black & blue coded surface albedo and what the relation between input data & pre-processing step color coding is.
R: We added legend for color coding explanation (Figure 1, Page 7, Line 182). Additionally, we changed surface albedo associated processes in blue for consistency. The color-coded data products under Data Acquisition were fed into the pre-processing associated with 1) surface setup in blue, 2) clouds setup in brown, and 3) providing ground truth in green.

L203f: Explicitly point out/summarize, what input is required by MCARaTS.
R: We added some description of MCARaTS inputs in the revised manuscript (Page 9, Line 245).

Tab1: Why is radiance limited to 2D output?
R: The radiance calculations are performed for a given altitude. Thus, the output is 2D (along x and y) *without* vertical dimension.

L238: I like your scene selection and the corresponding reasoning a lot. However, the results are not (yet?) analyzed and discussed explicitly for those diverse conditions.  Maybe a few summarizing word on performance depending on the different surface & cloud types could be added?
R: We added some discussion (Page 26, Line 742 and Page 40, Line 1157) in the revised manuscript about the challenges associated with the complexity of the surface.

L290f: Clarify relation of surface reflectance and surface albedo; how is (RT-input?) albedo converted from (observed) reflectance? Does it imply an assumption on surface reflection type (Lambertian?).
R: The surface reflectance is directly used as surface albedo input to the RTM assuming a Lambertian surface. We added clarification in the manuscript (see Page 14, Line 372).

L310ff: Are 10m winds really a good proxy for cloud altitude winds? What about using AMV for the wind correction?
R: 10m wind might indeed not be the optimal choice for accurately accounting for the cloud movements. Using Atmospheric Motion Vectors (AMV) is a good idea for better representing the cloud movements in the wind correction step. Since we only aimed at providing a bare-bones structure of approaches in this paper and 10m wind speed data was readily available from the OCO-2 data archive, we performed the wind correction using the 10m wind speed only. In the future, we will fine-tune each approach (e.g., improve cloud detection algorithm, improve parallax correction and wind correction etc.) to get into a detailed inter-comparison between observations and simulations. Actually, a few members within our group are conducting such development, e.g., cloud detection in the Arctic, and the work will be published in the near-future.

Fig3: What from are the fine white lines in the figure (eg at lon ~ -108.0 between lat ~ 37.4-37.6)? Correctional shift artifacts? Are those indeed treated as clear-sky then?
**R:** The white lines (cracks) are artifacts from the parallax correction. Yes, those were treated as clear-sky in the RTM. These white cracks can be avoided if we 1) turn off the parallax correction; 2) coarsen the study domain (for example, from 250m in the manuscript to 500m); or 3) improve parallax correction by taking adjacent pixels into account. In this version of EaR$^3$T, 2) or 3) have not been implemented.

L330: Why only a slope is fitted, not an offset, too? Looking at Fig4b it seems to me, a steeper slope with negative bias (passing through the two maxima) would fit the data better. Do you have any hypothesis on the origin of the two occurrence maxima? Would a surface type dependent fitting/transformation possibly be better?
**R:** If with offset, when surface albedo of x channel is 0 (e.g., a pure dark scene), the y channel will arrive at offset as surface albedo for the same pure dark scene, which is not physically reasonable, especially when the offset is negative. The following Figure shows the location of OCO-2 measured surface reflectance and collocated surface reflectance from MODIS surface product. The data is divided into two categories – OCO-2 surface radiance greater than 0.23 (upper maximum) in red and OCO-2 surface reflectance smaller-equal to 0.23 (lower maxima). We assume that such patterns might be associated with imperfect cloud filtering in 8-day MODIS surface reflectance product (based on our experience). The surface parameterization (linear regression) we developed is an imperfect but working solution to arrive at satellite radiance simulations. In the continuation work of OCO-2 as mentioned in the manuscript, we can improve surface reflectance parameterization by, e.g., using MODIS surface albedo product and performing surface type dependent fitting/transformation.

[Figure]

Figure 4-extra: The same as Figure 2 in the manuscript except for circles indicating the location of OCO-2 surface reflectance observations with larger values in red (> 0.23) and lower values (≤ 0.23) in blue.

L332: Are the same fitted-a values used for the other two channels, or do you just refer to the same fitting procedure?

R: The same fitting procedure can be applied for other OCO-2 channels – in this paper, only a channel of Oxygen-A band of three OCO-2 bands (Oxygen-A, stong-$CO_2$, weak-$CO_2$) is discussed. We decided to remove the text to avoid confusion.

L440: I don't get that sentence. Does "each wavelength" here refer to monochromatic wavelengths? Or individual channels? What are the "hundreds of individual absorption coefficient profiles"? why are they spectrally spaced for "each wavelength"?

R: The reviewer is correct. We changed the text as follows to clarify: "For each OCO-2 spectrometer wavelength within a given channel, hundreds of …" (Page 19, Line 541). In other words, there are three levels here: (1) OCO-2 channel (Oxygen A-Band, Weak CO2 band, Strong CO2 band), (2) spectrometer-resolution wavelengths in each band, (3) ABSCO line-resolving spectral spacing. For each of the spectrometer-resolution wavelengths (2), hundreds of wavelengths from ABSCO (3) need to be considered. After the calculations are done, all those individual calculations are convolved using the ILS.

L462: Again, what is the "target wavelength"? A channel?

R: The reviewer is correct. The "target wavelength" here means a channel. We added the text "(this could either be an individual SSFR or a MODIS channel)" (Page 20, Line 574) for clarification.

L502f: Could you shortly mention, how the parallelization can be applied (incl. install requirements & how to control the use of the parallelization) and whether the APPs in their current setup use this feature?

R: There are two kinds of parallelization that can be achieved – from RT solver side and from EaR³T side. The RT-solver-side parallelization requires additional software libraries (e.g., OpenMPI) during the installation setup. The parallelization of EaR³T is done through multi-processing, thus is natively supported (with EaR³T itself). Yes, the APPs (in their current setup) does use parallelization by default. We clarified this in the newly added Appendix A in the revised manuscript (Page 40, Line 1140).

Fig5: Why is this done on latitude-averaged rather than on footprint base? As the text argues with it (for cloudy locations; L544), could you (also) provide an equivalent plot on footprint base? A difference plot could be helpful, too. Why is there no spread/uncertainty shading on the IPA results? What is the range of SZA in the observations within the plot range?

R: We did latitude-averaging because the footprints of OCO-2 are not orthogonal to the latitude/longitude. We provided the OCO-2 vs simulation (IPA) intercomparison over the specified domain at footprint base and added it as Figure 5b in the revised manuscript. Additionally, we added the uncertainty shading for the IPA simulations. The SZA of the observations ranges from 32.59° to 34.92° with an average of 33.57°.

Fig6: Again, what is the range of SZA in the observations over the scene? What is the errors/uncertainties introduced by using a fixed SZA in the simulations?

**R:** The solar zenith angle (SZA) of the observations over the scene ranges from 32.43° to 36.46° with an average of 34.42°. To evaluate the errors/uncertainties, we performed the simulation at 32.43° ($SZA_{Min}$) and 36.46° ($SZA_{Max}$) in addition to 34.42° ($SZA_{Mean}$). Figure 6-extra-2 (the following figure) shows the reflectance (hemispherically integrated radiance normalized by cosine of SZA) of (a) $SZA_{Min}$ vs $SZA_{Mean}$ and (b) $SZA_{Max}$ vs $SZA_{Mean}$. Linear regression (y=ax) is performed and shown in black line. For $SZA_{Min}$ vs $SZA_{Mean}$, the fitted slope is 0.995. For $SZA_{Max}$ vs $SZA_{Mean}$, the fitted slope is 1.002. In other words, we are getting errors on the order of <0.5% in the domain average from the variability of the SZA throughout the area. The one-sigma errors of both are minimal – 0.000045. Thus we think the errors/uncertainties introduced by SZA in the simulations are minimal.

[Figure]

Figure 6-extra-2: **(a)** Simulated Reflectance at SZA=32.43° ($SZA_{Min}$) vs Simulated Reflectance at SZA=34.42° ($SZA_{Mean}$). **(b)** Simulated Reflectance at SZA=36.46° ($SZA_{Max}$) vs Simulated Reflectance at SZA=34.42° ($SZA_{Mean}$).

L555: "This commonly known problem" - please add references.
**R:** Reference is added (Barker and Liu 1995; Page 24, Line 686).

L581ff: How can you be sure the simulation bias is (only or mainly) due to COT? What about effects of further forward model errors (e.g. errors in Reff, PFCT, surface reflection model)
**R:** We are aware that the 3D effects can affect both cloud optical thickness (COT) and cloud effective radius (Reff; Zhang et al., 2012; Fu et al., 2022). Reff affects the reflectance in two ways: 1) modulating absorption in the shortwave infrared; 2) determining the scattering phase function. Since the simulations is performed for a visible wavelength (650 nm), where cloud do not absorb, the first effect is minimal. The second effect is small because the phase function does not vary significantly with Reff relative to the impact of COT. Therefore, we believe that the reflectance 3D effect is dominated by the cloud optical thickness (COT) distribution is reasonable. From Figure 7 of the manuscript, we can see that a good agreement is achieved for generally low radiance, indicating surface is well-modeled in the simulation.

L607ff: Please add (again?) what the domain size (Nx, Ny, Nz) of this simulation is.
**R:** The 3D atmosphere of clouds has the dimension of [Nx=1188, Ny=1188, Nz=26] with resolution of [dx=0.25km, dy=0.25km, dz=0.5km], which leads to a (1188, 1188) output of radiance field. We added the clarification in the revised manuscript (Page 26, Line 744).

L644f: What kind of interpolation is used?
**R:** The used interpolation is linear interpolation (added clarification in Page 27, Line 782).

L669f: Why does a low-biased AHI COT introduce a high bias in IPA-based simulations here, in contrast to APP1?
**R:** For APP1, the comparison is made for reflectance whereas for AHI, the comparison is made for transmittance. In APP1, the low bias in reflectance indicates low bias in COT. High bias in transmittance, however, indicates low bias in COT (high transmittance is associated with optically thin clouds).

L662f & 700f: Shouldn't the comparison rather be made on the resolution of the input data, ie AHI resolution, ie integrating/averaging SPN-S data? how does it look if that is done?
**R:** Originally, we thought about two approaches of a) interpolating AHI simulations (3D irradiance field based on AHI cloud retrievals) into aircraft location, and b) integrating/averaging SPN-S data into AHI resolution. We selected method (a) over method (b) because AHI resolution changes (finest at nadir), thus method (a) was easier to implement. We think both methods produce the same results in histograms. As an example, we performed the analysis (integrating/averaging SPN-S data onto AHI resolution) for the flight track shown in Figure 8 of this paper, as indicated in Figure 8-extra (the following figure). First, we evaluated the interpolation effect for the simulations. The red and blue are irradiance simulations linearly interpolated at flight locations, whereas the magenta and cyan shows irradiance simulations closest to the flight locations. From the histograms, we can see the interpolations effects are minimal – no major shifts in histogram despite interpolation offers more continuity. Next, we sampled (binned) the SPN-S data at AHI resolution (gray). After doing that, we saw that the variability decreased (maximum and minimum values) but the histogram shape remained fairly similar.

[Figure]

[Figure]

Figure 8-extra: **(a)** The same as Figure 8 in the manuscript except adding SPN-S averaged over AHI resolution grids (in gray), irradiance calculations closest to the flight locations (magenta and cyan) instead of linear interpolation (red and blue). **(b)** Histograms (probability density).

L722f: "We found that the bias [...] is partially caused by the coarse imager resolution" - in my understanding of the manuscript, that is rather a hypothesis. have you tested that somehow (eg by comparing on an AHI resolution basis, averaging/integrating the SPN-S obs to the AHI resolution)?
R: We agree. We added some discussions (Page 30, Line 867) in the revised manuscript. As we mentioned in the response to previous comment, averaging the SPN-S observations to the AHI resolution will not change the distribution of the histogram.

L872ff: "bias [...] was either due" - reformulate. The bias is quite surely not exclusively due to only one of these (as "either" implicates). Those two are very likely two main contributors (as stated above, I think, regarding the role of coarse resolution, this remains a hypothesis so far as in my understanding you have not demonstrated that yet.
R: We meant to state the bias is due to a combination of the two – 1) coarse imagery resolution, and 2) 3D effects. We changed the wording to clarify in the revised manuscript (Page 38, Line 1096).

L686: "their distributions are completely different" - I do not agree at all. Apart from the Trans>1 tail (and the equivalent higher peak in IPA at Trans~0.9, they are fairly similar. Particularly when compared to the observations.
R: We removed "completely" in the revised manuscript (Page 29, Line 838).

Fig9: For my feeling(!), the observations have a surprisingly high amount of Trans>1 (I'd by-eye-guess ~20% of all data). Is that expected, ie is the phenomenon that common?
R: We agreed that the transmittance greater than 1 is in surprisingly frequent, but that is expected. During the CAMP$^2$Ex mission, we saw extremely variable cloud conditions – fast-evolving, cirrus above, ubiquitous small size cumulus humilis clouds (cannot be resolved by geostationary satellite imagery). The phenomenon is expected because of the 3D effects associated with the cumulus ("popcorn") clouds. One point to clarify is that the imagery from the geostationary satellite as shown in Figure 8a does not represent the general cloud conditions for the entire campaign, whereas the histogram provided in Figure 9 does use *all* the below-clouds measurements from the entire CAMP$^2$Ex, not just the flight track shown in Figure 8. To make this point transparent to the reader, we published "flight videos" that we created for most of the research flights during CAMP$^2$Ex in the revised manuscript. This provides a better understanding of the cloud conditions.

Or could other things contribute (like calibration)?
R: The reviewer is correct, the calibration uncertainty can indeed contribute. To evaluate how this can affect the results, we applied a scale factor of 0.93 to the SPN-S measurements, assuming SPN-S consistently took measurements with largest reported calibration uncertainty of 7%. This scaling, however, will result in a 7% low bias of the SPN-S measurements compared to calculations for the high legs when the aircraft flew at altitudes around 6 – 7 km. We are therefore rather confident that SPN-S's calibration is actually accurate. Still, it is worthwhile looking at the results when scaling SPN-S down to the lower end of its uncertainty range. This is shown in Figure 9-extra below. The down-scaled SPN-S measurements below clouds (green) do show better agreement with the calculations (red) in terms of 1) average values (dashed lines), and 2) the high transmittance peak of the histogram (thin clouds and clear-sky). However, at the low transmittance side of the histogram (below a transmittance of ~0.5, thick clouds), the agreement becomes worse after rescaling, and generally the measured shape of the histogram still differs significantly from that of the calculations. To summarize, the calibration uncertainties can be propagated into the histogram and may affect not only the average values, but also the histogram shape. However, adjusting the calibration within the uncertainty will only lead to better agreement of one aspect at the expense of another. In other words, down-scaling the SPN-S measurements will lead to a better agreement of the campaign-average transmittance, but lead to a low-bias of the measurements above the clouds, and lead to worse agreement on the low-transmittance end below clouds.

[Figure]

Figure 9-extra: The same as Figure 9 in the manuscript except removing IPA calculations (originally in blue) and adding downscaled SPN-S observations (by a factor of 0.93) in green.

Sec6: I'd like to see some characterizing stats of the CNN, e.g. histograms of COT and/or radiance of training and validation data and retrieval uncertainties for either data subset.

**R:** We provided some brief descriptions for the CNN in Appendix B (Page 43, Line 1174). The histograms of the COT of the training dataset is provided at Figure 10b in Nataraja et al., 2022 (also see the following figure).

[Figure]

Figure 10b in Nataraja et al., 2022: Histogram of the cloud optical thickness for the sampled dataset used for the CNN training.

L739ff: Does that mean that for more global/operational application, separate CNNs for different SZA(-ranges) would need to be trained (SAA could still be handled by automatized rotation, I assume)? what further work do you see for a more global application?

R: 3D bias mitigation towards global/operational application is indeed an over-arching, but somewhat distant, goal. This paper is just an initial step in offering tools and barebone structures of demo for approaching this problem. We are developing more sophisticated CNN algorithms – testing different CNN techniques, training CNN on various cloud and surface types under different solar conditions, and gradually extend the application to global/operational application. For example, a former postdoc in our group has extended such an application to geostationary satellite imagery. This work is in manuscript stage and will be submitted to AMT in the near-future. Aside from developing more sophisticated and comprehensive CNN algorithms, we are also developing techniques to further accelerate the 3D calculations such as GPU application, hierarchical gridding, etc. so that it becomes operationally feasible for 1) generating large dataset for training CNN and 2) achieving the radiance self-consistency concept proposed in the manuscript at a global scale.

Fig10: Add indicators of the direction of the sun (or of north) in (a). Remove colored dots in (a) and their mentioning in caption when not otherwise explained and used (did i miss that in text?). Is the (6.4km)2 area the one inside the black or the red rectangle in (b), or the circular area? Does the circular are in (b) correspond to the yellow-circled area in (a)? "Inside [...] shown instead" - Formulate more straightforward to ease understanding: Inside the rectangle = regridded obs, circular area outside the rectangle = observed red channel radiance at native image resolution? why is the regridding done specifically to 100m resolution and how? caption refers to solid black lines indicated square area - I can't spot that; missing or badly visible?

R: Figure 10a and 10b are modified (see the following Figure as well as in the revised manuscript) – Figure 10a: removed colored dots and added arrows to indicate true north, flight direction, and sun position; Figure 10b: removed the original red lines and changed the black lines into red for better distinction. After modification, now in the new figures, the red circle in Figure 10a indicates the circular area in Figure 10b; the $6.4\times6.4$ km$^2$ region is indicated by red lines in Figure 10b. The boxed area in Figure 10b is the study region, which is gridded into 100 m resolution to match the spatial resolution of the training dataset of CNN. The outside-box area in Figure 10b is shown with native camera resolution but is not used in the study. We modified the text and added clarification (Page 32, Line 924).

[Figure]

**Figure 10. (a)** RGB imagery of nadir-viewing all-sky camera deployed during CAMP2Ex for a cloud scene centered at [123.392°E, 15.2744°N] over the Philippine Sea at 02:10:06 UTC on 5 October, 2019. The arrows indicate the true north (green), flight direction (blue), and illumination (where the sunlight comes from, yellow). **(b)** Red channel radiance measured by the camera for the circular area indicated by the red circle in (a). Red squared region shows gridded radiance with a pixel size of 64x64 and spatial resolution of 100 m.

Fig10, Fig12: Please make sure, figure, caption, and text are consistent. Eg. text mentions 12 edge pixels for Fig12, caption 7 on each side.
**R:** Thank you for noticing the typo. The number of excluded edge pixels should be 7. We corrected the text in the revised manuscript (Page 33, Line 964).

Fig12: Would be nice to have the observation data (red rectangle region) repeated here in the same style as the simulations for easier comparison.
**R:** We added the measured radiance (Figure 12c) in the same format as simulations for easy comparison (Page 34, Line 990).

Fig13: Reconsider the colors used. black dots on low histogram value colors are badly discernible. why are there dots outside the histrogram colored area? why are there no dots in the low-radiance corner, but histogram colors indicate high density? Is that an interpolation effect of the plotting (better to plot original binned data w/o interplation/smoothing)? Are the data shown from native imagery or regridded image resolution (or mixed???)?
**R:** We changed the colormap to better distinguish black dots from background color. The dots outside the colored area is due to the color range assigned to the number of occurrence – before, only occurrence greater than 2 were picked up from the background color. We removed the background where no data exists for Figure 13 for better visualization. The low-radiance corner indicates clear-sky to very-thin clouds. Since the surface albedo of the ocean at 600 nm, although very dark, is not zero. Additionally, due to 3D effects (clouds scatter radiation to the clear-sky region), we rarely have a radiance of 0 in neither measurements nor simulations. The interpolation/smoothing is in the visualization but does not have an effect on the distribution. The black dots are the re-gridded data (100m resolution) indicated in Figure 12 and the colored heatmap is the 2D histogram of the data.

Fig14 & discussion (L821-824): First, together with the ambiguous plotting of the mean CRE lines in Fig14a, the formulation of the "minor finding" is not fully clear. Does it mean that meanCRE(COT_CNN) is similar for both IPA and 3D RT simulations, and the same applies for meanCRE(COT_IPA)? I.e. mean of black solid is similar to mean of dashed blue on the one hand and solid red similar to dashed green? That would be in agreement with the following concluding sentence (L823f) and at least with the mean indicating lines in Fig14b (Fig14a I can't judge since the overlap of the lines renders the colors indistinguishable). While I buy that for Fig14b (ie CRE above clouds), from by-eye-judgement, I doubt that for Fig14a (CRE below clouds), since the blue dashed compared to the black solid curve seems to be shifted towards lower CRE (ie I expect mean(blue) < mean(black)) while green dashed compared to solid red seems shifted towards higher CRE (ie I expect mean(green) > mean(red)). Furthermore, in Fig14a both black solid and dashed green on the one hand as well as red solid and blue dashed on the other are very similar to each other (ie the two IPA and the two 3D RT simulations, respectively, regardless of COT retrieval method) - in both shape (that is, regardless of any other issues, regarding below-cloud CRE I dont not agree with your L823 statement that the PDFs are very dissimilar) and location along the CRE axis. Is there anything wrong, maybe, with the color coding of either the PDFs or the mean indication lines? Please check color coding and your respective conclusions.

**R:** Thank you for your comments. We modified the Figure 14 to make the different colored lines more distinguishable (we changed the black lines into gray lines and increased the line thickness). Yes, we meant $\overline{CRE_{IPA}}(\overline{COT_{CNN}}) \approx \overline{CRE_{3D}}(\overline{COT_{CNN}})$ (blue dashed line overlay gray solid line) and $\overline{CRE_{IPA}}(COT_{IPA}) \approx \overline{CRE_{3D}}(COT_{IPA})$ (green dashed line overlay red solid line). We added clarification in the revised manuscript (Page 36, Line 1037). We understood the conclusion is counter-intuitive but can be explained. For example, for Figure 14a, let's stick with $CRE_{3D}(COT_{CNN})$ (gray) and $CRE_{IPA}(COT_{CNN})$ (blue). The $CRE_{IPA}(COT_{CNN})$ (blue) has a more symmetric PDF distribution, thus the mean (blue dashed line) is located at the middle of the PDF. The $CRE_{3D}(COT_{CNN})$ (gray), however, has a gamma-like distribution – most of the CRE are very-negative, thus the mean (gray solid line) locates closer to the where the PDF is peaked. Thus, even though the histogram looks shifted to the right from blue to gray, their averages are similar.

L904: Over what data is that median taken? The processed scene? Does that work satisfactorily, e.g. over bright surfaces like deserts or snow covered areas?
**R:** The median is taken over the processed scene. It works satisfactorily for the given scene but will undoubtedly fail for scenes with less contrast, e.g., clouds over bright surfaces in the Arctic. Again, the applications demonstrated in this paper aim at providing bare-bone structure of approach for addressing well-known problems in 3D radiation science. A few more robust and universal cloud detection/mask algorithms are under development within our group, e.g., utilize multi-angle observations and/or information of scene variance in addition to scene brightness. Such algorithms will be discussed in future papers by Yu-Wen Chen (graduate student in our group)

et al., which applies the EaR$^3$T to improve $CO_2$ retrievals for OCO-2 (extended work from App. 1).

**Technical corrections**

**C:** Fig1: Make the figure larger (e.g., in landscape orientation) to make it better readable. Add color coding info to the caption.
**R:** We modified the figure into landscape orientation and added legend for color coding (Page 7, Line 182).

**C:** L203: "includes Monte" -> "includes the Monte"
**R:** We changed to "includes MCARaTS" (Page 9, Line 244).

**C:** L267: Are only channels 1 and 2 of MODIS L1B product used? Please specify their wavelengths here.
**R:** Yes, you are correct. We added wavelength information for clarification (see Page 13, Line 330).

**C:** L313: A reference to the correction effect figure in Appendix B would be useful.
**R:** Reference added (see Page 14, Line 394).

**C:** L486f: I am unsure what the documentation sentence here is supposed to tell the reader. Maybe it's just that the "only" is out of place here? However, there no other mention of (further) documentation elsewhere in the text, so this seems odd here.
**R:** We changed the wording and moved the sentence to the end of Section 2.1 (see Page 11, Line 282).

**C:** L531: "scale the MYD09 field" - Please add, which parameter this refers to (surface albedo/reflectance, I assume).
**R:** Yes, you are correct, we added "surface reflectance" for clarification (see Page 22, Line 651).

**C:** Fig8a: The flight track line is hardly visible; make it thicker. Caption misses an explanation what the thin and thick line sections are.
**R:** We made the lines thicker for better visibility and added explanation for the thin and thick lines (see Page 29, Line 824).

**C:** Fig9: "Vice versa for the green" - Misleading formulation. Like for yellow, PD(obs)>PD(sim) here, not PD(obs)<PD(sim) as "vice versa" implies.
**R:** We changed the wording to make it rigorous (see Page 30, Line 848).

**C:** L699: "the simulation histogram peaks" -> "the simulation histograms peak"
**R:** Corrected (see Page 30, Line 853).

**C:** L728: "use a high-resolution imagery" - remove "a"
**R:** Corrected (see Page 31, Line 894).

**C:** L801f: "By contrast" -> "In contrast"
**R:** Corrected (see Page 35, Line 1013).

**C:** Fig14: Please use larger font in legend for improved readability. Find a better way to indicate the overlapping mean-value lines - it's indiscernible which of the dashed lines lies where, particularly in (a).
**R:** We increased the font size for the legend. Additionally, we changed black lines to gray lines for better distinction (see Page 36, Line 1045).

**C:** L887: "introduce a warming bias" - I rather suggest "warm bias"; with CRE<0 the term "warming" feels odd.
**R:** We adapted the change (see Page 38, Line 1110).

**C:** L946: "in the black box" -> rather "in the inset"? (equivalently at L952)
**R:** We adapted the change (see Page 46, Line 1238 and Page 47, 1262).

**C:** L1093ff: Add info where to you intend to submit Schmidt et al., 2022 (same Special Issue?)
**R:** Schmidt et al. 2022 will be submitted AMT but not to the CAMP$^2$Ex special issue. We added clarification for the reference (… to be submitted to Atmos. Meas. Tech. …) in the revised manuscript (Page 54, Line 1475).

| Geographical Region | Specified at Line 668: **extent** | Specified at Line 68: **region** | Variable (depends on aircraft location) | N/A | N/A |
| Z Grid (Number of Grids/Resolution) | 40 / 0.5 km Specified at Line 547: **levels** | 40 / 0.5 km Specified at Line 422: **levels** | 20 / 1 km Specified at Line 184: **levels** | 40 / 0.5 km Specified at Line 192: **levels** | 20 / 1km Specified at Line 197: **levels** |
| Wavelength | 770 nm Specified at Line 785: **wavelength** | 650 nm Specified at Line 70: **wavelength** | 745 nm Specified at Line 443: **wavelength** | 600 nm Specified at Line 57: **wavelength** | 600 nm Specified at Line 62: **wvl0** |
| Atmospheric Gas Profile | US standard atmosphere Specified at Line 549: **atm0** | US standard atmosphere Specified at Line 424: **atm0** | US standard atmosphere Specified at Line 186: **atm0** | US standard atmosphere Specified at Line 194: **atm0** | US standard atmosphere Specified at Line 200: **atm0** |
| Atmospheric Gas Absorption | Case specific Specified at Line 557: **abs0** | Default Absorption Database (Coddington et al., 2008) Specified at Line 431: **abs0** | Default Absorption Database (Coddington et al., 2008) Specified at Line 192: **abs0** | Default Absorption Database (Coddington et al., 2008) Specified at Line 201: **abs0** | Default Absorption Database (Coddington et al., 2008) Specified at Line 202: **abs0** |
| Cloud Top Height | From MODIS L2 cloud product Specified at Line 306: **cth_2d_l2** And Line 592: **cld0** | From MODIS L2 cloud product Specified at Line 280: **cth_2d_l2** And Line 466: **cld0** | From AHI L2 cloud product Specified at Line 211: **cth_2d** And Lines 215: **cld0** | 2 km Specified at Line 217: **cth** And Lines 217: **cld0** | From LES Specified at Line 205: **cld0** |
| Cloud Geometrical Thickness | 1 km Specified at Line 592: **cgt** | 1 km And Line 466: **cgt** | 1 km Specified at Line 215: **cgt** | 1 km Specified at Line 217: **cgt** | From LES Specified at Line 205: **cld0** |
| Cloud Optical Thickness | Two-Stream Approximation for MODIS L1B Reflectance at 250 m resolution Specified at Line 402: **cot_2d_l1b** And Line 592: **cld0** | Two-Stream Approximation for MODIS L1B Reflectance at 250 m resolution Specified at Line 337: **cot_2d_l1b** And Line 466: **cld0** | From AHI L2 cloud product Specified at Line 201: **cot_2d** And Lines 215: **cld0** | Two-Stream Approximation and CNN for camera red channel radiance/reflectance at 100 m resolution Specified at Lines 285 and 324: **cot_ipa** and **cot_wei** And Lines 217: **cld0** | From LES Specified at Line 205: **cld0** |
| Cloud Effective Radius | From MODIS L2 Cloud Product Specified at Line 313: **cer_2d_l2** | From MODIS L2 Cloud Product Specified at Line 287: **cer_2d_l2** | From AHI L2 cloud product Specified at Line 202: **cer_2d** | 12 micron Specified at Lines 285 and 380: | From LES Specified at Line 205: **cld0** |

| | | | | | |
|---|---|---|---|---|---|
| | And Line 592: **cld0** | And Line 466: **cld0** | And Lines 215: **cld0** | **cer_ipa** and **cer_2d** And Lines 217: **cld0** | |
| Scattering Phase Function | Mie

Specified at Line 598: **pha0** And Line 630: **sca** | Mie

Specified at Line 472: **pha0** And Line 504: **sca** | Mie

Specified at Line 222: **pha0** And Line 240: **sca** | Henyey-Greenstein (g=0.85)

Implicitly specified by default at Line 232: **mcarats_ng**

Notes: Lines 207, 208, and 237 can be uncommented (meanwhile commenting out Line 209) to turn on Mie | Henyey-Greenstein (g=0.85)

Implicitly specified by default at Line 221: **mcarats_ng** |
| Surface Albedo | From MODIS Surface Reflectance product and scaled by OCO-2

Specified at Line 520: **oco_sfc_alb_2d** And Line 629: **sfc_2d** | From MODIS Surface Reflectance product

Specified at Line 395: **mod_sfc_alb_2d** And Line 503: **sfc_2d** | 0.03

Implicitly specified by default at Line 237: **mcarats_ng** | 0.03

Specified at Line 236: **surface_albedo** | 0

Specified at Line 227: **surface_albedo** |
| Solar Zenith Angle | From OCO-2 geolocation file

Specified at Line 615: **sza** And Line 633: **solar_zenith_angle** | From MODIS geolocation file

Specified at Line 489: **sza** And Line 507: **solar_zenith_angle** | Variable (depends on aircraft location and date and time) | 28.90°

Specified at Line 352: **geometry['sza']** And Line 240: **solar_zenith_angle** | 29.16°

Specified at Line 228: **solar_zenith_angle** |
| Solar Azimuth Angle | From OCO-2 geolocation file

Specified at Line 616: **saa** And Line 634: **solar_azimuth_angle** | From MODIS geolocation file

Specified at Line 490: **saa** And Line 508: **solar_azimuth_angle** | Variable (depends on aircraft location and date and time) | 296.83°

Specified at Line 353: **geometry['saa']** And Line 241: **solar_azimuth_angle** | 296.83°

Specified at Line 229: **solar_azimuth_angle** |
| Sensor Altitude | 705 km (satellite altitude)

Implicitly specified by default at Line 625: **mcarats_ng** | 705 km (satellite altitude)

Implicitly specified by default at Line 499: **mcarats_ng** | N/A, three-dimensional irradiance outputs at user-defined Z grid | 5.48 km (flight altitude)

Specified at Line 354: **geometry['alt']** And Line 242: **sensor_altitude** | 705 km (satellite altitude)

Implicitly specified by default at Line 221: **mcarats_ng** |
| Sensor Zenith Angle | From OCO-2 geolocation file

Specified at Line 617: **vza** And Line 635: **sensor_zenith_angle** | From MODIS geolocation file

Specified at Line 491: **vza** And Line 509: **sensor_zenith_angle** | 0° (nadir)

Implicitly specified by default at Line 237: **mcarats_ng** | 0° (nadir)

Implicitly specified by default at Line 232: **mcarats_ng** | 0° (nadir)

Specified at Line 230: **sensor_zenith_angle** |
| Sensor Azimuth Angle | From OCO-2 geolocation file

Specified at Line 618: **vaa** | From MODIS geolocation file

Specified at Line 492: **vaa** | 0° (insignificant for nadir) | 0° (insignificant for nadir) | 0° (insignificant for nadir)

Specified at Line 231: |

| | | | | | |
|---|---|---|---|---|---|
| | And Line 636: `sensor_azimuth_angle` | And Line 510: `sensor_azimuth_angle` | Implicitly specified by default at Line 237: `mcarats_ng` | Implicitly specified by default at Line 232: `mcarats_ng` | `sensor_azimuth_angle` |
| Number of Photons | $1\times10^8$ per run

Specified at Line 72: `_photon_sim`
And Line 640: `photons` | $1\times10^8$ per run

Specified at Line 71: `_photon_sim`
And Line 514: `photons` | $1\times10^7$ per run

Specified at Line 56: `photon_sim`
And Line 246: `photons` | $1\times10^8$ per run

Specified at Line 56: `_photon_sim`
And Line 246: `photons` | $1\times10^8$ per run

Specified at Line 66: `photon_sim`
And Line 234: `photons` |
| Number of Runs | 3

Specified at Line 638: `Nrun` | 3

Specified at Line 512: `Nrun` | 3

Specified at Line 245: `Nrun` | 3

Specified at Line 244: `Nrun` | 3

Specified at Line 233: `Nrun` |
| Mode (3D or IPA) | 3D and IPA

Specified at Line 786: `solver`
And Line 641: `solver` | 3D

Specified at Line 620: `solver`
And Line 515: `solver` | 3D and IPA

Specified at Lines 380 and 381: `solver`
And Line 247: `solver` | 3D

Specified at Lines 391 and 392: `solver`
And Line 247: `solver` | 3D

Specified at Line 210: `solver`
And Line 236: `solver` |
| Parallelization Mode | Python multi-processing

Specified at Line 643: `mp_mode` | Python multi-processing

Specified at Line 517: `mp_mode` | Python multi-processing

Specified at Line 250: `mp_mode` | Python multi-processing

Specified at Line 249: `mp_mode` | Python multi-processing

Specified at Line 238: `mp_mode` |
| Number of CPUs | 8

Specified at Line 642: `Ncpu` | 8

Specified at Line 516: `Ncpu` | 16

Specified at Line 314: `Ncpu`
And Line 249: `Ncpu` | 12

Specified at Line 248: `Ncpu` | 24 on clusters

Specified at Line 237: `Ncpu` |

[revised manuscript text omitted]

Longitude correction (positive from west to east):

$$\delta lon = \frac{u \cdot \delta t}{\pi \cdot R_{Earth}} \times 180° \qquad (B3)$$

Latitude correction (positive from south to north):

$$\delta lat = \frac{v \cdot \delta t}{\pi \cdot R_{Earth}} \times 180° \qquad (B4)$$

where $u$ and $v$ are the domain-averaged 10 m zonal and meridional wind speeds, and $\delta t$ is the time difference between two different satellites that fly on the same orbit. Figure A2 shows the cloud location after applying the parallax (Appendix D1) and wind correction for the cloud field in the inset from Figure 2.

[revised manuscript text omitted]

**Page 22: [1] Deleted**         **Hong Chen**         **12/6/22 5:16:00 PM**

---

## Author Response (AR2)

**Response to Reviewer #1**

Title: The Education and Research 3D Radiative Transfer Toolbox (EaR$^3$T) – Towards the mitigation of 3D Bias in Airborne and Spaceborne Passive Imagery Cloud Retrievals

Authors: Chen et al.

Recommendation:
Major revision

Summary:
The authors of this manuscript developed a modularized Python package EaR$^3$T which automates the process of 3D radiative transfer calculation. They illustrated the broad range of applications of this 3D-RT package by showing four examples of 3D radiance simulation and cloud retrievals.

The work is solid and requires tremendous effort. The package developed by the authors is also very useful and has significant potential for 3D-RT-related applications.

However, I found that the current structure of this manuscript is difficult to follow. This is because there are too many low-level technical details in the first 4 sections, which could severely distract readers from the primary scientific findings of this manuscript (See my major comments).

Also, there are a few places where more explanations and clarifications are needed (see my major comments). Considering all this, I recommend a major revision for this paper.

**R:** Thank you for your comments.

**Major comments:**

Comments on the paper structure:
1. Though I understand that it takes tremendous effort to develop this package, I found the description in Section 2.1 Overview (especially L180-229 and Table 1) is too technical (not very scientific-related). It is better to move this part into an appendix and focus on the four applications showing the advantage of using 3D RT model in radiance simulation and cloud retrievals, the scientific part of this manuscript.

**R:** Thank you for your comments. We agree and moved the tables and technical descriptions about the input and output parameters into Appendix A1 (Page 35, Line 870).

2. The same comments to the Section "3. EaR$^3$T Procedures" (L400-434), L488-509. You can create a separate manual for your package, but in the text, you might want to focus on the scientific part. Only discuss the input/output sources for your applications. For example, some description of Table 2 is good enough.

**R:** We modified Section 3 – 1) the code walk-through in Section 3 was moved into Appendix A2 (Page 39, Line 919) and 2) the procedure description originally in Section 2 was moved into Section 3 (Page 16, Line 398). At the end of Section 3, we cross referenced Appendix A.

**Comments on the results:**
L541: Figure 5. It might be better to indicate the cloud and clear-sky regions in Figure 5. Also, show the RGB image here for more easy pairing your results with the cloud and clear-sky areas of the RGB image. For the EaR$^3$T IPA calculation, what is your input? Column gas and temperature profiles or still 3-D gas and temperature fields?

**R:** We added Figure 5b (Page 18, Line 463) in the same format as Figure 2 but for monochromatic IPA radiance calculations to provide context of the entire domain. For both IPA and 3D radiance calculations, the input column gas and temperature profiles are 1D profile, which only contains vertical variability and assumes atmospheric gases and thermodynamic parameters are horizontally homogeneous. All the radiation related input parameters are now provided in Appendix A1 (Table A1, Page 38, Line 906) for transparency. 3D atmospheric gas and temperature fields are supported but not used in any of the applications shown in the paper. We plan to build support for longwave in the near-future, where 3D temperature field will play an important role in determining 3D cloud radiative effects in the longwave.

L544: "In the cloudy regions", where are exactly cloudy regions? It seems to be not mentioned in the previous context.
**R:** We added text clarification (see Page 19, Line 470) as well as Figure 5b (newly added, Page 18, Line 463) for providing context for the domain.

L534: In this context, you attributed the biased clear-sky 3D-RT radiance bias to the surface reflectance (red). But could the diffuse radiance from the nearby clouds contribute to your biased simulations at the clear-sky regions? If true, how do you determine which components contribute more to the bias radiance?
**R:** What you described is exactly what we tried to explain in the paper. 3D effects contain two parts – brightening of clear-sky regions (or optically very thin clouds) near clouds as they are net photon "recipients" and darkening clouds themselves (if clouds are optically thick) as they are net photon "donors". At the OCO-2 footprints of the domain, the model input surface albedo is directly taken from the OCO-2 retrieved surface reflectance under clear-sky conditions without considering the brightening 3D effects. Since we saw an increase of the radiance from IPA to 3D calculations (see more explanation in our response to the next comment), we argue that the OCO-2 derived surface reflectance is indeed too high because of the cloud vicinity effect. One potentially confounding factor that we did not consider in our manuscript is that we did not include aerosols that can alter the results. To your question about whether we can determine which component contributes more, the answer is no we cannot with the simple showcase in the paper. However, it is possible if we extend this case to multi-spectral and multi-angle and even with sub-orbital observations from aircraft. Our next following paper will use this strategy for approaching radiation closure.

I would suggest doing the following experiment: conduct a simulation over a large clear-sky region (so no diffuse radiance from nearby clouds) to see if the clear-sky 3D-RT still

overestimates the radiance. If so, you can attribute the bias to the surface reflectance. Now, I cannot determine this because I'm not sure how large is your clear-sky region in Figure 5.

R: A new simulation is not needed as we have IPA calculations (blue in Figure 5a). From the IPA calculations, we can see that within the clear-sky regions (e.g., latitude range of [38.05°, 38.3°]), the radiance simulations are roughly in agreement with the observations. As stated in the previous response, the bias can be traced back to the surface reflectance contaminated by 3D cloud radiative effects in the raw OCO-2 observations (referred to as stage 1), which were used for surface reflectance retrieval without any correction for the 3D effects. Next, we perform 3D calculations (red) and the radiance simulation goes even higher than the observations (stage 2), indicating an enhanced 3D effects when radiation is allowed to scatter from clouds into clear-sky. Such increase high bias from stage 1 to stage 2 corroborates our assumption that the bias resides in the surface reflectance when it was contaminated by 3D effects at stage 1. Of course, the aerosols, which are not considered in the RT, can play an important role and potentially alter the results. Thus, we changed our wording by adding "probably" (Page 18, Line 454).

L582-584: "Since the MODIS reflectance is not self-consistent…of COT". Here you have implicitly assumed that Era3T calculation is the truth. It would be good to discuss the input of IPA calculation, especially the different input components to EaR$^3$T IPA and a standard plane-parallel 1D RT model, since those different components contribute to the simulation difference here. It will be self-consistent if you use the same plane-parallel 1D RT model to retrieve those cloud parameters.

R: The ground truth we are relying on is the MODIS (and OCO-2) observed radiance, not EaR$^3$T's radiance calculations. We agree that oversimplifying 1D RT (for example, when using the two-stream approximation as done in the original version of this paper), or using erroneous inputs to either 1D or 3D calculations can introduce errors. What we mean by radiance self-consistency is the following: We first map the radiance to cloud products (COT etc.) in a very similar manner as done in the heritage retrieval, except that it leads to products at a higher spatial resolution than provided in the operational L2 product (matching the resolution of the L1B radiances). In the first step, we then run forward calculations from these retrievals with 1D RT (EaR$^3$T-IPA), and compare these calculations with the original radiance observations. Provided that no systematic errors were made in these calculations, they should agree with the original observations because IPA is essentially the inverse operation to the original retrieval. Examples of these calculations are shown in Figure A4a and A4b (newly added, Page 47, Line 1107). Any deviation from the 1:1 line here indicates errors in the input cloud or surface properties, or errors in the radiative transfer itself. In the case of Figure A4a, for example, the calculated reflectances on the lower end (clear sky) are sometimes lower than the measurements. However, overall, the scatter around the 1:1 line is negligible. In sum, when using IPA as the forward model, the calculated radiances are largely in agreement with the observations as they should be. However, in reality, 3D-RT is at work in nature. If the retrieved L2 properties are correct, then it should be the radiances derived from forward calculations in 3D-RT, not 1D-RT that reproduce the measured radiances. MODIS (or OCO-2) radiances are not self-consistent when 3D-RT does not reproduce the original observations. Lack of self-consistency can be attributed to two factors: (1) 3D effects as described (2) any errors in the input fields that were already detected by comparing IPA calculations with the observations. The 3D effects can be isolated by comparing them

against the IPA baseline. In our examples (Figure 7 and 13b against the newly added IPA baseline in Figures A4a and b), the 3D effect dominates the radiance inconsistency by far. The 3D bias in radiances (Figure 7) can be as large as 40% for optically thick clouds (reflectance greater than 0.3), whereas there is no systematic bias in the IPA (Figure A4a). Thus, using the radiance self-consistency to evaluate 3D biases is justifiable.

L790: Figure 13 shows you are using a Two-stream approximation. Have you tried larger stream numbers (at least 4 streams)? For your CNN trained on EaR$^3$T-based 3D Radiance field, how many streams do you use? It will be a fair comparison only if these two use the same number of streams.

**R:** Thank you for your comments. We agreed that two-stream approximation can lead to artifacts in the 3D effects we demonstrated in the paper as it is only a good proxy for irradiance but not radiance. To address this, we changed the IPA cloud retrieval method from two-stream approximation to IPA reflectance to COT (cloud optical thickness) mapping (described in Appendix C2, Page 45, Line 1073) obtained from the same radiative transfer process we used in the paper. This way the IPA consistency is ensured, and any biases exist in the 3D radiance self-consistency check stem from 3D effects. The two-stream approximation (or higher stream) is simplified analytical solution for 1D-RT, which uses plane-parallel assumption and independent pixel approximation. While the 3D radiance simulations from EaR$^3$T use MCARaTS as 3D-RT solver, which uses Monte-Carlo method to output simulation results based on photon statistics, no plane-parallel assumption or independent pixel approximation is involved, which better depicts the reality of nature than the 1D-RT. There were a few simplifications we made in our preliminary CNN, e.g., surface albedo of 0, Henyey-Greenstein phase function (g=0.85) for clouds etc. (details are discussed in the revised manuscript as well as Nataraja et al., 2022) that we plan to improve in the near-future. This will be elaborated more in the upcoming publications. Nevertheless, even this CNN out-performs the heritage IPA retrieval.

Minor comments:

L41: "In contrast to isolated case studies in the past, EaR$^3$T… irradiance.": This claim seems misleading. Even the RT calculations with the plane-parallel RT models are made independently for each pixel, they are usually verified against over large regions as long as we have observations. Also, they are verified against multiple sources. I would recommend deleting this claim from your abstract.

**R:** We agree with this statement. However, we are not criticizing the plane-parallel approach here, nor are we questioning that many studies do a large amount of data aggregation. What we are trying to express here is that the automation capability allows us to perform massive RT 3D calculations for an entire campaign as opposed to limited case studies in the past (e.g., from our own papers) that often focused on single legs. For this reason, we decided to keep this statement. We did delete "isolated" to make this clear (see Page 2, Line 42).

L77: "Once the CNNs are trained": remove "the"

**R:** We think "the" should be here as "CNNs" was described in previous sentence and here we are referring to that specific "CNNs".

L107: "cloud fields with minimal user input": Please rephrase this sentence. It would be better to emphasize how it automates the whole 3D-RT calculations instead of saying "with minimal user input". Readers can have different interpretations of "With minimal user input". One unpleasant interpretation is to use this tool as a black box.

**R:** We rephrased the statement to the following (also see Page 5, Line 126):

"It can be operated in two ways– 1) with minimal user input, where certain RT parameters are bypassed through default settings, for quick radiation conceptual analysis; 2) with detailed RT parameters setup by user for radiation closure purpose."

L148-149: "The code, along…": You can move this sentence to the Section of "Data and Code Availability", as required by EGU journals.

**R:** We moved the text to the Data and Code Availability Section (See Page 50, Line 1162).

L248: Figure 2: Are those circles representing OCO-2 showing their actual spatial resolution, or just an illustration, since those footprints in the figure are not distorted and shown as circles?

**R:** The circles can only indicate the location of the OCO-2 footprints, NOT the spatial resolution. We added clarification in the caption of Figure 2 (See Page 10, Line 244).

L306-307: Move the data source of OCO-2 data to the Section of "Data and Code Availability"

**R:** We moved the OCO-2 data source to the Data and Code Availability Section (see Page 50, Line 1162).

L388: "Only radiance data from the red channel were used in this paper.": Is there any reason why you only use observations from 626nm band?

**R:** Yes, the reason is the CNN model used in this paper (App. 4) was trained on synthetic data with realistic radiance simulated at 600 nm. We selected the red channel of the camera since the wavelength (centered at ~626 nm) is close to what CNN was trained.

L410: Again, move the link to the Section of "Data and Code Availability"
**R:** We moved the links to the Data and Code Availability Section (see Page 50, Line 1162).

L478: "In addition to MCARaTS, planned solvers…": Move this part to the "Summary and Conclusion" Section as future work.
**R:** We moved the text to the conclusion (Page 34, Line 862).

L607: "For technical references,": change to "for the running time of simulation," or something more specific.
**R:** Changed (see Page 22, Line 545).

**Response to Reviewer #2**

I am very satisfied with the authors' response to the reviews, both with the changes to the manuscript as well as the clarifications and explanations in the response documents.

**R:** Thank you for your constructive comments and your time and effort dedicated to the review. After we revised our manuscript to address your comments along with comments from another reviewer during the initial review process, we received a third review during the technical correction process that requires some additional changes including text modification and result update. Please note that the manuscript was therefore changed beyond the recommendations you made, described below.

Text modification:
- The table for EaR$^3$T output parameters (originally in Section 2) has been moved to Appendix A1;
- Some descriptions for EaR$^3$T procedures (originally in Section 2) have been moved to Section 3;
- The code walk-through for App. 1&2 (originally in Section 3) has been moved to Appendix A2 (newly created, Page 39, Line 919);
- The cloud detection method has been polished (Appendix C1, Page 44, Line 1036);
- Appendix C2 (Page 45, Line 1073) has been updated to the new IPA reflectance-to-COT (cloud optical thickness) mapping method;
- The parallax correction has been polished – now includes a "cloud crack" treatment (Appendix D1, Page 47, Line 1113).

Result update:
- We switched from the usage of surface reflectance provided by MYD09A1 to white-sky albedo provided by MCD43A3 for surface albedo parameterization due to the finding of MCD43A3 is more reliable;
- We updated the IPA method for retrieving COT$_{IPA}$ based on cloud reflectance;
- Figures 3 – 7 and 11 – 14 have been updated.

I have only very few very minor comments:

- I'd appreciate if a cross-ref to Appendix A (not only to TabA.1 therein) is added in the manuscript (somewhere appropriate Sec1-3)
**R:** We cross-referenced Appendix A at the end of Section 3 (Page 17, L434).

- Sec2: with addition of Appendix B, the limitation "four of which are discussed in this paper" (L174) seems not necessary anymore (and contradictory to the following 5-items list). Also it would be better to use Apps1-4 or panels (a)-(d) from Fig1 instead of "the first four [applications]". L207f is redundant to the 5-items list above. There's no fifth column in Fig1 (L216) anymore; rather refer to it as panel (e).
**R:** We removed the obsolete text and changed the "fifth column" to "panel e" (Page 9, Line 217).

- Fix typesetting of formula on L733.
R: Corrected (Page 25,  Line 627).

- Fig9: rather "blacked filled" (or similar) to refer to SPN-S (I was at first stupidly looking for a black line...)
R: Corrected (Page 25, Line 636).

- Fig12: maybe adjust colorbar range or colorbar map to more clearly pronounce the IPA-CNN diffs again. They were clearer in previous Fig version, now get somewhat lost on the benefit of diffs to the obs (I appreciate the addition of the obs very much, though!)
R: We changed the colormap (Page 29, Line 730).

- L970: one of the App1-5 is redundant.
R: Corrected (Page 35, Line 872).

- Fig6 discussion: Could be worth to add to the manuscript (from response to Reviewer #2) that errors from fixed-SZA assumption are negligible.
R: Added (Page 21, Line 532).

- Fig14 discussion: Could be worth to add to the manuscript the explanation of the seemingly counterintuitive conclusion (as given in response to Reviewer #2)
R: Added (Page 26, Line 660, and Page 31, Line 775).